# Automatic detection of cell-cycle stages using recurrent neural networks

**Abin Jose** [1]*, **Rijo Roy** [1], **Daniel Moreno-Andrés** [2], **Johannes Stegmaier** [1]*

**1** Institute of Imaging and Computer Vision, RWTH Aachen University, Aachen, Germany, **2** Institute of Biochemistry and Molecular Cell Biology, Medical School, RWTH Aachen University, Aachen, Germany

☉ These authors contributed equally to this work.
* * abin.jose@lfb.rwth-aachen.de (AJ); johannes.stegmaier@lfb.rwth-aachen.de (JS)

## Abstract

Mitosis is the process by which eukaryotic cells divide to produce two similar daughter cells with identical genetic material. Research into the process of mitosis is therefore of critical importance both for the basic understanding of cell biology and for the clinical approach to manifold pathologies resulting from its malfunctioning, including cancer. In this paper, we propose an approach to study mitotic progression automatically using deep learning. We used neural networks to predict different mitosis stages. We extracted video sequences of cells undergoing division and trained a Recurrent Neural Network (RNN) to extract image features. The use of RNN enabled better extraction of features. The RNN-based approach gave better performance compared to classifier based feature extraction methods which do not use time information. Evaluation of precision, recall, and F-score indicates the superiority of the proposed model compared to the baseline. To study the loss in performance due to confusion between adjacent classes, we plotted the confusion matrix as well. In addition, we visualized the feature space to understand why RNNs are better at classifying the mitosis stages than other classifier models, which indicated the formation of strong clusters for the different classes, clearly confirming the advantage of the proposed RNN-based approach.

## Introduction

Chromatin segregation errors during mitosis are a source of chromosomal instability and a hallmark of diverse pathologies [1, 2], including cancer [3, 4]. Therefore studying in detail the highly dynamic process of mitosis, as well as the fate of chromosomes during cell division is of great importance for both basic and clinical research. The study of mitotic cytology and its pathological phenotypes date back more than a century with the pioneering work of Walther Flemming [5]. However, the quantitative study of morphological parameters in fixed samples [6–8] or time-lapse images of living cells have recently emerged as a powerful tool to elucidate underlying molecular mechanisms [9–11]. In the pursuit of less biased and time-consuming quantitative approaches, an immense effort is being made to generalise algorithms capable of automatically segmenting, tracking, extracting and quantifying different feature aspects of the mitotic events in time-lapse microscopy image sequences. A sequence indicates a series of

**Data Availability Statement:** The data underlying the results presented in this study are in location: https://osf.io/8qdgm/files/osfstorage For information about data please refer to: (https://osf.io/b6gy5/wiki/home/).

**Funding:** This work is supported by internal funds.

**Competing interests:** The authors have declared that no competing interests exist.

frames. Such tools for unbiased and comprehensive cytological analysis reflecting natural and pathological changes in different phases of the cell-cycle are urgently needed in both basic and translational research [12]. The technological advancement in the area of machine learning with deep neural networks can potentially support biological and clinical researchers. For example, these automated methods could give a faster and more precise analysis of cell behaviors under different drug treatments [13, 14].

## Related work

The state-of-the-art methods identifying cell-cycle stages in time-lapse microscopy records extract image features and then expert biologists train machine learning algorithms to generate cytological classifiers. For example, CellCognition [10] is a technique that is applied for annotating time-resolved mitosis stages from live cell image sequences. The high similarity between images in some stages of mitosis and smooth transition will introduce high classification noise at the state transitions. CellCognition demonstrates that the incorporation of time information into the annotation can compress the classification noise and reduce the confusion between images of similar morphology. In this approach, an object detection method is implemented initially to identify the location of each cell. They used local adaptive thresholding [15] with watershed split-and-merge error correction [16] to detect individual cells with high accuracy. Then for each object, they calculated features describing texture and shape. From these features, they combined a supervised machine learning technique for classification with a Hidden Markov Model (HMM) to reduce misclassification by incorporating time information. Even though this method gives good performance, some of the mitosis stages are wrongly classified. This is due to the high degree of similarity between some classes as well as the fact that some classes have fewer training samples than others. They assumed that each state of the cell at a given time point depends on the previous state and considered an HMM [17] for the error correction. All the parameters of the HMM like prior probabilities, transition probabilities, and prediction probabilities are derived automatically from the output of a Support Vector Machine classifier. Then they derived the overall maximum likelihood path for the sequence by the Viterbi algorithm [18]. HMM increased the overall accuracy of the model by eliminating the misclassifications at stage transitions.

In 2012, [19] proposed an unsupervised method for identifying the stages of mitosis from the images captured using time-lapse microscopy. This paper introduces a clustering algorithm based on a temporally-constrained combinatorial clustering (TC3) method as a module in the CellCognition [10] software. This approach uses the features extracted from the time-lapse microscopy images of human tissue culture cells (HeLa 'Kyoto' cells). This method classifies the cells into interphase and five stages of mitosis, which are prophase, prometaphase, metaphase, anaphase, and telophase. The CellCognition software calculates the synchronized time series of cell features based on the shape and texture of the tracked cells over time. The authors then convert these features to a lower data dimension using principal component analysis [20] (PCA). They use this converted data as the starting point of the TC3 algorithm. For each cell trajectory, the TC3 algorithm will cluster temporally liked features to a user-defined number of classes. The authors first used a binary clustering algorithm to divide the PCA features into 3 clusters based on mitotic subgraph properties. Further, they used the TC3 algorithm to cluster into subclasses. The authors also state that the performance of the model can be increased by using TC3 results to initialize a Gaussian Mixture Model (GMM). Also, further extended the performance by extending this GMM model results to HMM. This model predicted the cell behaviors closer to the user annotations. For each subcluster, the TC3 algorithm will do an exhaustive search with all possible combinations of labels. Within each set, it

calculates the distance measures of features assigned to the same class of labels and selects the set of labels with the least distance measure as the final output. TC3 algorithm on the set of features used for this dataset gives results similar to the user annotations.

LiveCellMiner [11] is an open-source tool developed to analyse live-cell time-lapse records obtained in different microscopy platforms in a quantitative and unbiased manner. This software tool can track single cell fates and extracts, analyses, and visualises biological relevant image features from 2D+t microscopy images. The study of human cells passing through mitosis under different experimental conditions has settled the proof of principal application of this tool. Some of the functionalities of this software include fully-automatic segmentation and tracking of the cells. This tool can be also used to extract the quantitative features of the cells being tracked. This tool is available as an extension package in the MATLAB toolbox SciXMiner [21]. Object detection functionalities of this software are utilized to locate different cell nuclei in an image containing many cells and then crop a square region around each cell. One of the datasets used in this paper is extracted by this tool. Segmentation labels for each cell are extracted by using a modified version of Otsu's method [22]. The cell trajectory synchronization application of LiveCellMiner is used to identify the duration of different mitotic stages. Using this tool mitosis as well as wrongly detected trajectories can be automatically identified. The default settings for mitotic analysis on LiveCellMiner divides cell-cycle into three classes: interphase, early mitosis (prophase to the end of metaphase), and late mitosis (anaphase, telophase and until G1 when the nucleus recovers to interphase). LiveCellMiner uses three different methods for synchronizing cell trajectories. The first method is a TC3 clustering method [19]. In this method, classical image features like area, circularity, and intensity of each cell are calculated and clusters with minimum within-class variance are extracted for detecting the interphase to prophase transition. The metaphase-to-anaphase transition comes from the tracking and uses a user-defined distance of sister chromatin masses to identify anaphase onset. The second approach is based on the first approach and additionally uses trainable LSTM [23] networks which evaluate each trajectory as a whole, identifying erroneous trajectories. The third method uses Convolutional Neural Networks (CNNs) features extracted by using GoogleNet and uses LSTMs to predict the state sequence for all time points. These predicted time points are post-processed with an HMM model which allows only valid state transitions. Then the most likely sequence of stages is identified using the Viterbi algorithm [18].

A Recurrent Neural Network (RNN) [24] is a type of deep learning technique used with temporal sequence data. Artificial neural networks such as CNNs are meant for single data points which are independent of each other. However, in sequential data or time series data, one data point is dependent on the previous data points. Since RNNs are temporally connected, they can store the information from prior inputs to influence the current input and output. Thus, the output of an RNN is dependent on not only the current time point but also on the previous time points. This type of network is commonly used in speech recognition and natural language processing. RNNs can also be used with convolutional layers to extend their application to video data. When the sequence data is very large, the gradient information during the training of an RNN is not able to propagate back to the earlier time points. This problem is called vanishing gradients. Cho et al. [25] proposed a type of RNN called Gated Recurrent Unit (GRU) to solve the vanishing gradient problem. GRU solves this problem by using an update gate and a reset gate, which decides what information has to be passed to the output.

In 2016, Ondruska et al. [26] proposed an RNN-based approach as 'Deep Tracking', for the end-to-end detection of objects from sensor data. In this paper, the authors used RNN networks to extract features from the input data. For the end-to-end object tracking directly from raw sensor data for robots, in 2016, Ondruska et al. [26] proposed the deep tracking approach.

The idea was to use raw sensor data without any feature engineering as the input and produce an output that also included the detection of the occluded objects. This paper also followed an unsupervised training approach to achieve this result. The authors stated that this is the first approach that uses unsupervised end-to-end tracking of objects using sensor data. Classical approaches at the time of this work solved object detection in separate stages, with an object detection stage and a tracking stage. The proposed deep tracking approach is trained end-to-end, thereby overcoming the hand-engineering required in such a separately executed model. This is achieved by exploiting the sequential model in the form of RNNs to learn complex dynamics from raw data to object tracks. In this model, the hidden states of the RNNs can capture certain appearances and motion patterns of objects. Inspired by the Bayesian filtering approach and considering a generative model, the authors stated that there exists a Markov process, which completely captures the state of the world. From the hidden state representation, the location of each object at each time frame is trained to predict using binary cross-entropy loss. In a later work, Ondruska et al. [27] proposed that each of the object classes can be learned from this hidden state distribution by using RNNs. The network can perform a semantic classification from the rich information learned in the hidden states. They proposed that this network can be trained end-to-end with a small amount of labeled data. This classification network is introduced once the deep tracking network is learned. This method outperformed many semantic classification methods available at that time. To be able to track moving objects throughout the time sequence, the network must remember the location and other properties of each object. To achieve this tracking, the authors used GRUs as the processing steps at each layer of the RNN. Convolutional GRUs are utilized to maintain the resolution. From the learned hidden layer, two convolutional networks are employed to predict the semantic segmentation as well as the class of the object. The network is trained with softmax loss for the classification and a binary cross-entropy loss for the segmentation.

## This paper

In this paper, we propose an approach that identifies different stages of a cell during mitosis using deep learning techniques from tracked cell images. The tracked cell images follow a single cell in different time frames and avoid background clutter from the surrounding cells. The aim is to find the phases of mitosis of the cell in different time frames. In other words, the aim is to find the temporal segmentation of a video sequence of cell data. This means that the class labels are assigned to each frame of the video sequence to classify the mitotic phases. Inspired by the model proposed by [27], we modify the network architecture and incorporate time-related propagation of features for better classification of mitosis stages. The main contributions of this paper are summarized here:

- We propose a novel network architecture, Time Encoded ResNet18 Model, which uses the GRUs to capture the inherent time dependency between different frames during mitosis, resulting in better classification.

- We compare the performance of the proposed RNN-based approaches to the original deep learning-based classification networks such as ResNet18 [28].

- Experiments were conducted on two different datasets: a) LiveCellMiner dataset [11], which contains images in three different phases of mitosis. b) Zhong et al.'s dataset which contains images in six different phases of mitosis.

- By visualizing the feature space using PCA [20], the effect of the incorporation of time information is studied.

- We also plotted the confusion matrices to identify the classes that cause confusion and identified potential areas for further improvement in classification accuracy.

- For quantitative evaluation, we have measured the precision, recall, and F-score.

- The reconstruction of the center cell using tracking network is visualized for the images in different mitotic phases.

## Proposed approach

Given a cell mitosis video sequence, it has become of utmost importance in the context of cell-biology research to find the temporal segmentation of cell-cycle stages. While there are several successful methods to track [29, 30] the cells from a video sequence, identification of time-dependent cell-cycle stages or cytological phenotypes using deep learning methods is still a challenging task. Most existing approaches [10, 11, 19] for cell stage classification start by extracting features separately for each of the frames and then try to identify the stage sequence by a similarity-based grouping of the feature vectors. However, the temporal dependency of features that were extracted from successive frames is not considered here. To incorporate this time dependency, using normal CNNs would not be sufficient. In this context, we propose an RNN-based approach to address the problem of time-related features. Since in RNN, the networks are connected in time, it can effectively transfer information across temporal data.

### Base model

Each cell follows a set of unique steps during cell division. Thus, the different phases of cell mitosis always occur in the same order and the current state can be considered as being dependent on its previous states. Therefore, a cell in any frame of a cell mitosis video sequence is dependent on the same cell from earlier frames. To this end, inspired by the networks proposed by [26, 27], we propose our first architecture by considering time-related features for the classification of the cell mitosis stages. The network design of the model is given in Fig 1. Our proposed model consists of four main layers:

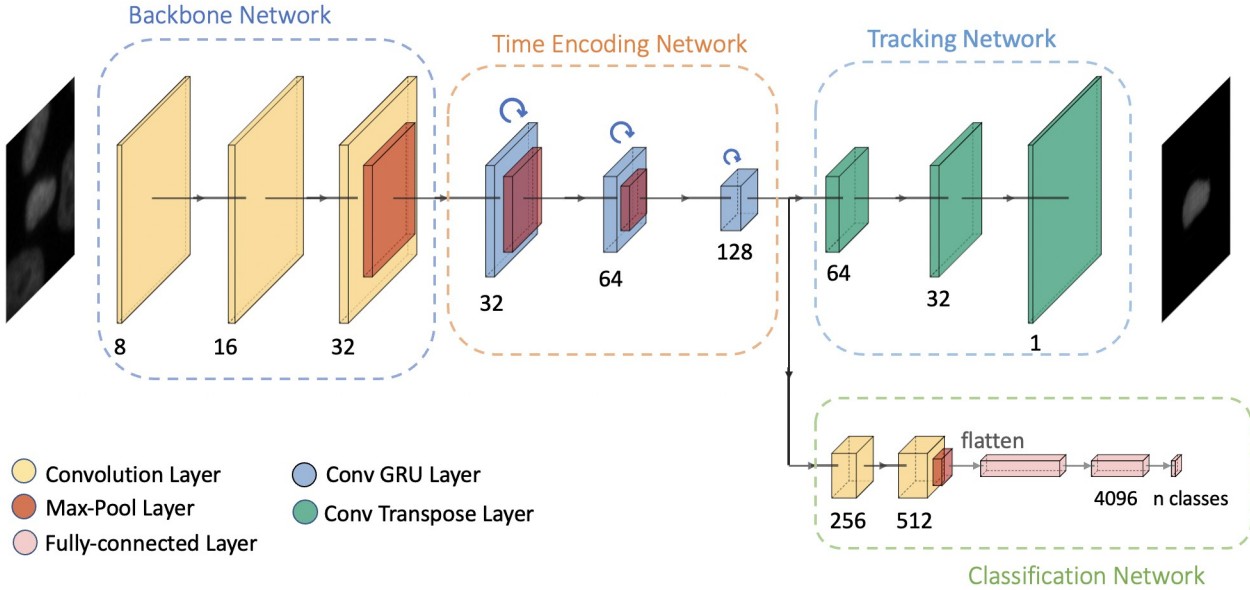

**Fig 1. Network architecture.** Illustration of the architecture of our proposed base model applied to the $n^{th}$ frame of a video sequence [31].

- A backbone network for feature extraction.

- A time encoding network to incorporate time information into the features.

- A tracking network to track the cell.

- A classification network for the mitosis stage prediction.

The initial results and experiments using this base model are summarized in our previous work [31]. The experiments show that adding the time information into the features, helps to better identify cell-cycle stages. Each of these modules is explained in detail in [31].

We further modified this baseline architecture, with a view that deeper networks can create deep feature representations with complex functions and are more efficient. The modified model, time encoded ResNet18, which we elaborate in this paper also has the aforementioned modules. The details of this model are explained in next subsection.

## Time encoded ResNet18

This model introduces deeper layers between each GRU layer by combining the architecture of ResNet18 [28] with RNN layers. Combining these two modules introduces a deeper architecture with convolutional layers between each RNN layer. This in turn enhances the feature extraction capability of the model. Combining these two modules helps to propagate the features extracted at different layers and increases the classification performance. RNN combines the features of the current frame with the information from the previous frames in the video. In a process like a cell mitosis, this encoding of time information helps in predicting the class to which the cell belongs, because one stage is dependent on the previous stages and always occurs in the same order. After the time encoded features are extracted, tracking and classification networks are used for reconstructing the center-cell and predicting the class of the cell respectively.

**Time encoded backbone network.** The main difference from the base model [31], is that there is no separate backbone and time encoding network. They are combined together into a single module, as shown in Fig 2. The time encoded backbone network extracts the features with time information for each frame in the video sequence. This is achieved by combining a state-of-the-art model like ResNet18 with the properties of an RNN. ResNet18 contains eighteen deep layers with eight residual blocks. The proposed network uses an architecture of ResNet18 with convolutional GRUs between the residual blocks to transfer time information between frames. Similar to the base model, this model has three convolutional GRUs. The GRU layers in this model have been positioned such that the dimensions of each GRU layer is different. This helps to propagate information at multiple scales. In the ResNet18 architecture, a max-pooling layer reduces the dimensionality after every two residual blocks. Thus, each convolutional GRU is placed between two residual blocks of the ResNet18 architecture. This addition of convolutional GRU layers does not change the dimensionality of the features of the ResNet18, but help in propagating time information by combining features from the previous time frames to the present frame. The channel sizes of the convolution GRU layers are 128, 256, and 512. The original grayscale image is rescaled and converted to an RGB image for matching the input requirements of the ResNet18 architecture. Starting with the initial image $x_{img} \in \mathbb{R}^{3 \times H_0 \times W_0}$ (RGB image), the time encoded backbone generates a lower resolution activation map $f \in \mathbb{R}^{C \times H \times W}$. Typical values used for channels and image resolution are $C = 512$, and $H,W = \frac{H_0}{32}, \frac{W_0}{32}$ respectively. The ResNet18 layers are loaded with pre-trained weights of network

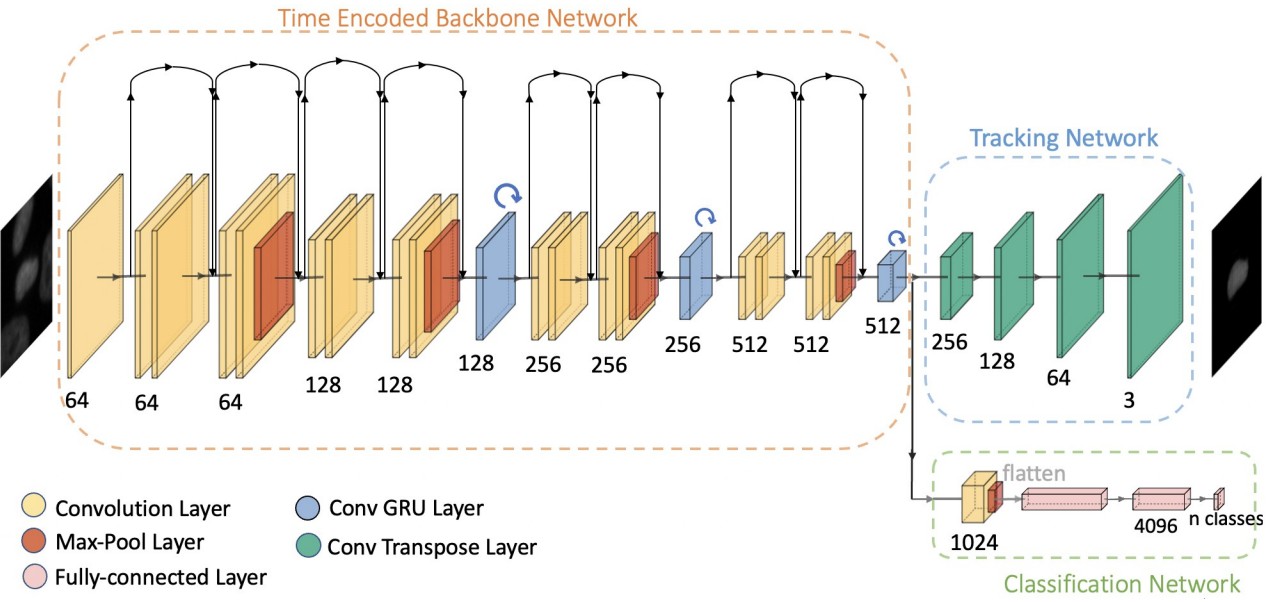

**Fig 2. Network architecture.** Illustration of the architecture of our proposed time encoded ResNet18 model applied to the $n^{th}$ frame of a video sequence where GRU blocks are connected between residual blocks of ResNet18.

trained on ImageNet dataset and the complete network is fine-tuned with the cell sequence dataset.

**Tracking network.** Since the time encoded backbone network has a deeper architecture compared to that of the base model, the output features are at a lower spatial dimension. So compared to the base model, which uses three transpose convolutional layers, this tracking network uses four transpose convolutional layers to reconstruct the center-cell to the original image dimension from the output of the time encoded backbone network. 3×3 kernels are used in all transpose convolutional layers. The typical values of the channel dimensions used in each transpose convolutional layer are 256, 128, 64, and 3. The final layer has an output with three channels matching the RGB image size with each channel representing the grayscale content. Thus a resolution equal to the RGB input image is reconstructed and the output dimensions are $x_{tracking} \in \mathbb{R}^{3 \times H_0 \times W_0}$.

**Classification network.** A shallow classification network is enough to predict the class from the output features of the time-encoding backbone network. A convolutional layer, a max-pool layer, and two fully-connected layers make up this network. The convolutional layer has 1024 filters and uses 3×3 kernels. The feature dimensions are then reduced by a 2×2 max-pooling layer. The features are flattened after the max-pooling layer. Then a series of two fully-connected networks is added that predicts the class to which each frame belongs. The classification network's last layer will provide an output whose size is equal to the number of classes.

## ResNet18 classifier

The two models that were discussed previously use RNN to propagate information between subsequent frames. To compare the performance of the models without using RNN, we propose to use a state-of-the-art deep learning classification model. This model consider each frame in the video as independent of other frames in the same sequence. This approach uses transfer learning on a ResNet18 [28] model. A pre-trained model architecture with the last

layer replaced with the number of classes is used. Then the complete network is trained with image datasets of the mitosis sequence. ResNet18 has eighteen deep layers with eight residual block connections. In the final layer, the number of outputs is updated with the number of classes in our dataset. Thus the network predicts the stage of mitosis for each input image. The complete network is then trained with the cell mitosis dataset. In this way, the network can learn more information related to microscopy images, whereas the pre-trained ResNet18 model is trained with non-microscopy images. The network consists of seventeen convolutional layers and one fully-connected layer. All the convolutional layers have a kernel size of 3×3. Eight residual blocks are implemented with two convolutional layers. A max-pooling operation reduces the dimension after every two residual blocks. Finally, the fully-connected layer predicts the class to which each image belongs.

## Training details

This section explains training with the supervised approaches proposed in this paper. The models are trained end-to-end. The proposed models that use GRUs to deal with time-based features have a tracking network and a classification network. The tracking network reconstructs only the center-cell out of many cells in the frame so that the feature space will contain information about this cell. The classification network predicts the stage of mitosis of the cell. The fully-connected network in the final layer of the classification network has an output dimension equal to the number of classes to predict. The losses for these two networks are combined simultaneously by using a weighting factor. The deep learning classification models, which do not have a GRU, have only the classification network that predicts the stage of the mitosis of a cell, and the loss is optimized during training. The following section explains how these models are trained in different scenarios.

### Training base model and time encoded ResNet18

Since time encoding networks are used to help the flow of information across time frames, each image in a sequence is dependent on previous images. Therefore in these models, the sequence of images belonging to the same cell is given as the input.

**Tracking network loss.** The input images used for training base model are grayscale images with a resolution of 96×96. Time encoded ResNet18 uses 3-channel RGB images as input and has a resolution of 224×224. Hence, the original images are converted to 3 channels and resized for training these models. The output of the tracking network in both models has a dimensionality equal to that of the input training images. The tracking network reconstructs only the center-cell from the input image. This is achieved by calculating the loss between the predicted output and a masked image. The masked image is the input image masked with the segmentation of the center-cell. The last layer of the tracking network uses a sigmoid activation layer. Then the loss between the predicted and the expected input is calculated using a binary cross-entropy loss [32] as given below:

$$L_{\text{track}} = -\frac{1}{W_0 \cdot H_0} \sum_{w \in W_0} \sum_{h \in H_0} (y(w, h) \cdot \log(\hat{y}(w, h)) + (1 - y(w, h)) \cdot \log(1 - \hat{y}(w, h))), \quad (1)$$

where $y(w, h)$ and $\hat{y}(w, h)$ are the input and predicted values of the network at pixel location $((w, h))$. $W_0$ and $H_0$ are the input image dimensions. The segmentation mask of new images can also be extracted using a simple intensity threshold operation on the predicted output of the tracking network.

**Classification network loss.** In the classification network, the final layer has a number of outputs equal to the number of mitosis stages. This layer has a sigmoid activation function.

The network is trained with a one-hot vector of the ground-truth values. This helps the network to train to predict the probability of the image belonging to each class. The training loss function used is binary cross-entropy loss [32] as shown below:

$$\mathrm{L}_{cls} = -\frac{1}{\mathrm{N}_{class}} \sum_{c \in \mathrm{N}_{class}} (y_c \cdot \log(\hat{y}_c) + (1 - y_c) \cdot \log(1 - \hat{y}_c)), \tag{2}$$

where $y_c$ is the one-hot embedding of the ground truth values and $\hat{y}_c$ is the output predicted values of the classification network for the $c^{\mathrm{th}}$ class. $N_{class}$ is the total number of classes in the dataset. The class with the highest probability is chosen as the predicted class during inference. The total loss is calculated as a weighted sum of the tracking and classification losses. A parameter $\lambda_{wt}$ is used to weigh the losses.

$$\mathrm{L}_{tot} = \mathrm{L}_{track} + \lambda_{wt} \cdot \mathrm{L}_{cls}. \tag{3}$$

### Training ResNet18 classifier

The deep learning architectures which used ResNet18 classifier has only the classification network loss. In these models, each image is considered to be independent of the previous image in the same sequence. The final layer of these networks has a number of outputs equal to the number of classes to predict. A one-hot embedding vector of the ground truth is used to train this network. This network is trained with the cross-entropy loss [32] to predict the probability of the image belonging to each class as shown below:

$$\mathrm{L}_{cls} = -\frac{1}{\mathrm{N}_{class}} \sum_{c \in \mathrm{N}_{class}} (y_c \cdot \log(\hat{y}_c)), \tag{4}$$

where $y_c$ is the one-hot embedding of the ground truth values and $\hat{y}_c$ is the classification network's output prediction values for the $c^{\mathrm{th}}$ class. $N_{class}$ is the total number of classes in the dataset. The class with the highest probability is chosen as the predicted class during inference. The total loss of models with deep learning architectures is the same as the classification loss as shown below:

$$\mathrm{L}_{tot} = \mathrm{L}_{cls}. \tag{5}$$

### Datasets

For experimental evaluation, we have used two main datasets. The first dataset is provided by Moreno-Andrés et al. [11] and the second dataset is provided by [19]. Both datasets contain microscopic image sequences of the mitotic process from human HeLa cells expressing H2B-mCherry as fluorescent chromatin marker. These images are acquired using time-lapse microscopy. In the following subsections, these two datasets are explained.

### LiveCellMiner dataset

As explained in, the LiveCellMiner [11] tool allows the analysis of mitotic phases in 2D+t microscopy images. These images contain human tissue culture cells as they undergo mitosis and are acquired using widefield and confocal microscopy. These acquired images have many cells in each time frame. The software tracks the position of each cell in the image and then extracts the tracked single cell image. The software then checks and eliminates cells that do not

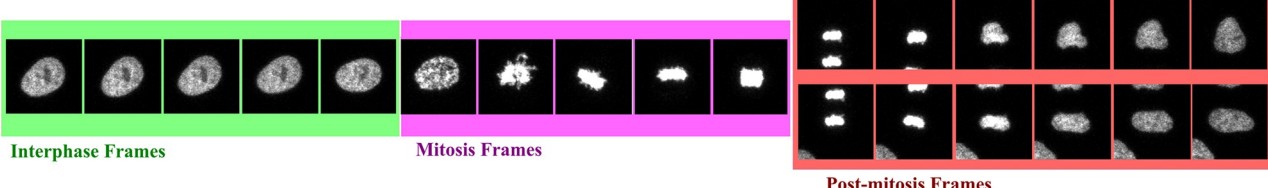

**Fig 3. Illustration of the images of a cell undergoing cell division.** Illustration of the images of a cell that undergoes cell division and also assigned to the three different cell-cycle classes: interphase, mitosis (prophase, prometaphase, metaphase), and post-mitosis (early and late anaphase and telophase) classes [11]. Two sequences are available after the cell-splitting—each follows one daughter cell after mitosis.

undergo cell division. A region is cropped around each tracked cell from each time frame if the cell undergoes cell division. Each of these cropped images has a resolution of 96 × 96 pixels, and the target cell is in the center of the cropped image. For each cell division sequence, 90 frames are available with this resolution. This dataset is divided into three classes. This software identifies interphase to early prophase transition and metaphase/early anaphase to late anaphase transition as reference patterns for the alignment of interphase or postmitotic frames, and then automatically detects interphase to prophase and metaphase to anaphase transitions. Then it divides the data into interphase, mitosis, and post-mitosis classes. A sequence of images belonging to different classes is illustrated in Fig 3. Experts corrected the predicted labels of the LiveCellMiner tool and released them as ground truth annotations. Along with this tool, four different image datasets are published. All four datasets are acquired on the human HeLa cells expressing H2B-mCherry transfected with indicated siRNA oligonucleotides in eight-well μ-slide chambers. These datasets contain images that were taken three minutes apart. The total number of training sequences in each of these datasets and a detailed description of these datasets are given below. The first dataset is the LSM710 dataset. In this dataset, the cells are imaged using an LSM710 confocal microscope (Zeiss) and ZEN software (Zeiss). 1458 sequences are available for training with the LSM710 dataset. The second dataset is the LSD1 dataset [33]. It was acquired using an LSM5 live confocal microscope (Zeiss) and ZEN software. With this dataset 1042 sequences are available for training. RecQL4 dataset [34] is the third dataset and was acquired using LSM5 live confocal microscope and ZEN software. 1214 sequences with this dataset are available for training. The last dataset is the NikonXLight dataset [35]. This dataset is imaged with a widefield module of a Ti2 Eclipse (Nikon) equipped with a LED light engine SpectraX and GFP/mCherry filter sets and using elements software (Nikon). This dataset contains 1229 sequences for training. Fig 4 illustrates a few images of these four datasets at different stages of cell splitting.

## Zhong et al.'s dataset

The dataset provided by Zhong et al. [19] also uses time-lapse microscopy images of human tissue culture cells (HeLa cells). This dataset contains cells as they undergo cell division. Each frame in the dataset is labeled as interphase or one of the five mitosis stages. Cell images belonging to these six stages, with their state diagram is shown in Fig 5. The labeling by different biologists can be inconsistent despite the well-defined chromatin morphology. The biologist's annotations were subjected to a dissimilarity study that showed minor inconsistencies between annotations made by the same person on different days, but significant differences between annotations made by different users. The gold standard for the labels is chosen by a majority vote among these user annotations. The dataset consists of seven image sequences with a total of 326 cell division events uniformly sampled with an interval of 4.6 minutes. Since

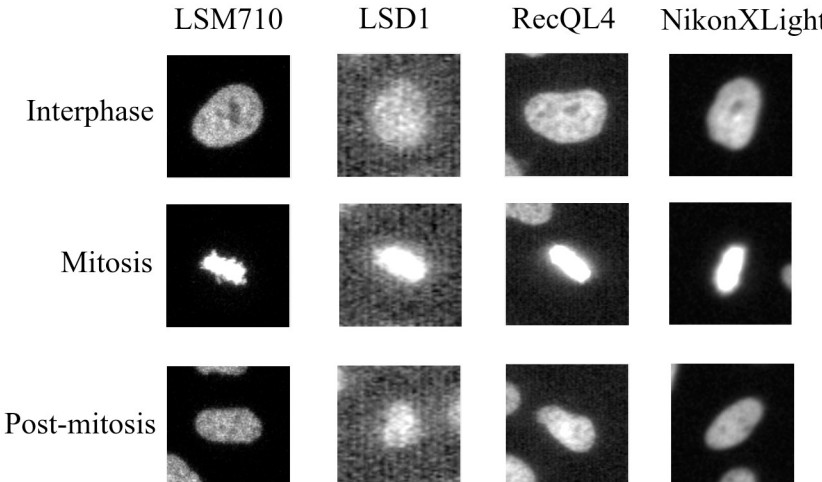

**Fig 4. Images from four different datasets.** Images belonging to four datasets of LiveCellMiner at different cell-cycle stages [11].

mitosis stages occur in a shorter time compared to the interphase stage, the distribution of cells in different classes is highly uneven. In this dataset, each image has a resolution of $96 \times 96$ pixels, and each cell sequence has a length of 40 frames.

## Evaluation criteria

This section explains the different evaluation techniques, used to compare the models. We compared our results with some of the state-of-the-art methods as well as between our different models. Here we measured classification performance such as classification accuracy, confusion matrix, and some of the measures which are derived from the confusion matrix. For the evaluation of multi-class classification, three possible cases for predictions are available for each class. These are true positives (TP), type I error or false positives (FP), and type II error or false negatives (FN) [36]. TPs of a class are the correctly classified data, whereas FPs and FNs are the incorrectly classified data. The data that belongs to this class but is predicted as one of the other classes is FN. FPs are the data that are predicted into this class that belongs to another class.

### Accuracy

Accuracy is defined as the sum of all TPs from all the classes divided by the total number of data points. Here in this paper, we calculate frame-to-frame accuracy from each sequence and

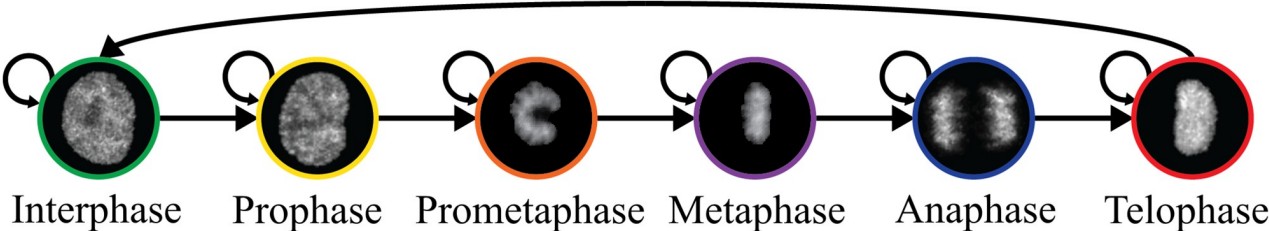

**Fig 5. Cell mitosis stages in the form of a state diagram.** Illustration of the different stages of cell-cycle [19] as a state diagram. Cell images belonging to different classes are labeled with different colors.

average it over the total length of data. The accuracy that is discussed in this paper is the average accuracy generated by evaluating the ground-truth labels and the predictions from the test dataset. A higher value of accuracy means better performance. The value is in the range of 0 to 1.

## Confusion matrix

The confusion matrix can visualize the three cases of prediction, that is TPs, FPs, and FNs. The TPs for each class lie on the diagonal of the confusion matrix [36]. The FPs and FNs for a class will be on the corresponding columns and rows respectively, such that the true class is on the y-axis and the predicted class is on the x-axis. In this paper, we present the normalized confusion matrix.

## Precision and recall

The precision represents the proportion of correct classified images within a class to the total number of images classified in this class. I"t is the ratio of TP to the sum of TP and FP as shown in Eq 6. A higher value of precision means better performance. Recall again indicates the ratio of TP to the sum of TP and FN.

$$\text{precision} = \frac{\text{TP}}{\text{TP} + \text{FP}} \qquad (6)$$

$$\text{recall} = \frac{\text{TP}}{\text{TP} + \text{FN}} \qquad (7)$$

## F-score

The F-score denotes the harmonic mean between precision and recall. The values of the F-score lie in the range of 0 to 1. A higher F-score means that the classification has a better performance. The F-score is defined as in Eq 8.

$$\text{F} - \text{score} = \frac{2.\text{precision}.\text{recall}}{\text{precision} + \text{recall}} \qquad (8)$$

## Experiments

In line with our previous work [31] we conduct the experiments for each of the five datasets mentioned in Section. We compared the performance of the proposed RNN-based methods with ResNet18 classifier. The classification models treat each image independently. Hence, the number of images in the batch equals the batch size. Models with GRU layers always consider each image in a sequence dependent on the previous images from the same sequence. Each sequence is used together as input during training to produce this dependency. So, in this case, the batch size equals the number of sequences. The experiments are carried out with these different models to evaluate their performance. In experiments, we observed that a batch size of two or four sequences performs better than a batch size of one. In this paper, the typical batch chosen has 4 sequences for the models with GRU layers. This is because, due to the graphic processing unit's memory limitations, some deeper models cannot employ batch sizes greater than 4. For the models using ResNet18 classifier, the batch size used is 64 images. For the

**Table 1. Hyperparamater values.** The hyperparameter values chosen for experiments after initial grid-search based tuning.

| Hyperparameter | Model | | |
| --- | --- | --- | --- |
| | Base model | Time encoded | Classifier |
| Batch Size | 4 Sequences | 4 Sequences | 64 Images |
| Learning Rate | 0.01 | 0.001 | 0.001 |
| Learning Rate Scheduler | 0.001 | 0.0001 | 0.0001 |
| Loss Regularization | 0.01 | 0.1 | - |
| Train-Test Ratio | 0.85 | 0.85 | 0.85 |

training of the base model where a pre-trained model is not available, a learning rate of 0.01 proved to provide better training than lower or higher values. The typical learning rate used for models with a pre-trained model is 0.001. Inspired by [27], the learning rate is scheduled to reduce to 10 percent of the current value after 2500 training iterations. The weighting parameter value controls the weighting between multiple losses. We observed that the weighting parameter value of 0.01 has better accuracy with the base model and a value of 0.1 with the time encoded ResNet18 model by using the basic grid search [37] approach. Besides the aforementioned hyperparameters, data augmentation is introduced in all experiments to increase the robustness and avoid overfitting. Rotations by multiples of 90 degrees and flips in vertical and horizontal directions are the data augmentations used. We also split the datasets into separate training and testing datasets. The typical value of the train-test ratio used is 0.85, such that 85% of the total data is used to train the models, and the remaining 15% is used as test dataset. 10% out of the test images were used as the validation set. The experiments were done for 20 repetitions and the standard deviation is measured. Table 1 shows the typical values of the hyperparameters used in the experiments, which were determined by grid search-based parameter tuning. The cells will grow back to the interphase stage after they undergo mitosis. The expert annotation of Zhong et al.'s dataset considers this, while the LiveCellMiner dataset does not consider this aspect. In LiveCellMiner datasets, the cells moving back to the interphase stage are annotated as the part of the post-mitosis class. Due to the resemblance between these cells in the post-mitosis class and those in the interphase class, this will pose issues during training. Studies from the LiveCellMiner [11] showed that the recovery back to the interphase stage happens around 15 to 20 frames after mitosis. To overcome the above issue, during training, we labeled all the frames after around 15 to 20 frames from the post-mitosis stage as the interphase class. During inference, the predicted interphase recovered class after the mitosis stage is assigned to the post-mitosis stage and evaluated. This also solves the problem of similarity between images for deep learning classification networks like ResNet18. Thus, the recovery back to interphase does not affect the prediction accuracies of these models.

## Experimental results

This section presents the results of the proposed approaches on different datasets. The code is available in the git link, https://github.com/Rijo756/cell-cycle-stages-identification. Accuracy, confusion matrix, precision, recall, and F-score are used to quantify the classification performance. The results are estimated on two models using time encoding with GRU layers and also on the ResNet18 classifier. The models are trained for 12 epochs with the LiveCellMiner dataset and 40 epochs with Zhong et al.'s dataset (both around 10, 000 batch iterations). We also plotted the label matrix which helps in comparing the class predictions of proposed models to the user annotations. Another experiment to study how the learned feature space looks

for different models is also plotted for different datasets by taking the first 3 principle components, to visualize how the feature space looks in a lower dimensional space.

### Results on LiveCellMiner dataset

In this section, the results are presented in four parts. Each part belongs to results from one of the four datasets available from LiveCellMiner.

**LSM710 dataset.** The results of the proposed models trained with the LSM710 dataset from LiveCellMiner are presented in this subsection. Fig 6 visualizes the label matrix of the ground truth annotation and the labels generated by the proposed models on 50 test sequences. The y-axis denotes these sequences, and the x-axis is the length of each sequence. Each of the four proposed models produced excellent results on this dataset. The performance of each of the predicted labels can be computed using the accuracy of the predictions.

Table 2 demonstrates the frame-to-frame accuracy of each of the proposed models. With the LSM710 dataset, the classification using the ResNet18 model got the least accuracy, and the time encoded ResNet18 model has the maximum performance in terms of accuracy. Both the proposed models have higher accuracy than the ResNet18 architecture. This indicates that introducing features from the previous time step using GRU layers seems to help increase the performance.

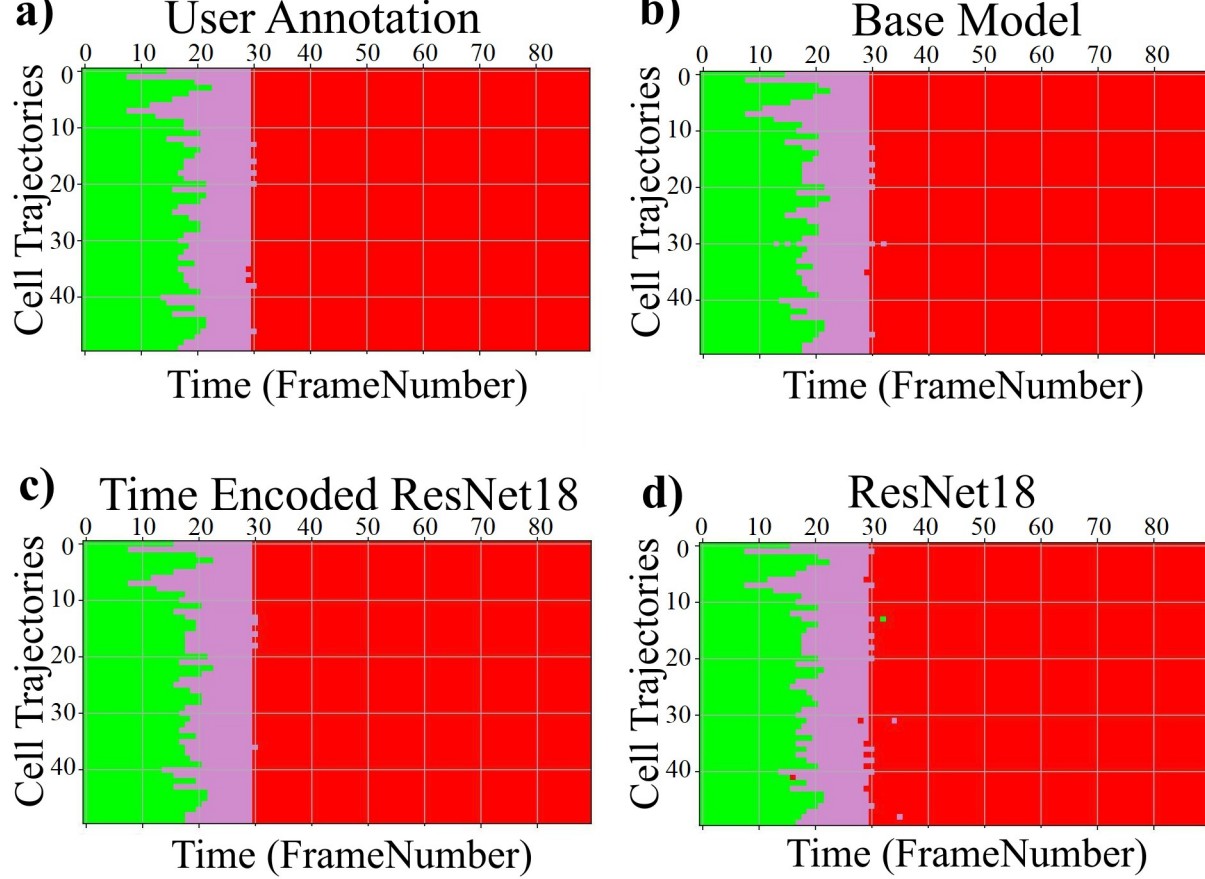

**Fig 6. Label matrices.** Label matrices of user annotation and the predictions by proposed models for 50 sequences selected from the test data of the LSM710 dataset. The y-axis represents different cell trajectories, and the x-axis represents the length of each trajectory. Green, magenta, and red represent the interphase, mitosis, and post-mitosis classes respectively.

**Table 2. The frame-to-frame accuracy values.** The frame-to-frame accuracy value of various proposed models using the LSM710 dataset. Boldface indicates the best performance.

| Model | Accuracy |
|---|---|
| LiveCellMiner | 99.39 |
| Base Model | 99.529±0.094 |
| Time Encoded ResNet18 | **99.565± 0.040** |
| ResNet18 | 99.347±0.039 |

The LSM710 datasets are labeled as interphase, mitosis, and post-mitosis stages. The confusion matrix represents the correct predictions as well as the wrong predictions. Fig 7 visualizes the normalized confusion matrices for the proposed models. It is evident from the confusion matrices that all the models were able to classify the images into correct stages in most of the cases as the values of the diagonal elements are higher. The precision, recall, and F-score of the three classes of various models are presented in Table 3. The F-score, precision, and recall are highest for the interphase and post-mitosis stages for the time encoded ResNet18 model. For the mitosis stage, recall and F-score for LiveCellMiner have slightly better performance but the time encoded ResNet18 also has comparable performance here as well.

Fig 8 visualizes the reconstructed output image from the tracking network. The base model and the time encoded ResNet18 model has a tracking network to reconstruct the center-cell from the input image. The ResNet18 model is a classification network and only outputs the stage each input image belongs to. Both models using a tracking network were able to reconstruct the center-cell. The reconstructed cell from the base model seems to be better compared

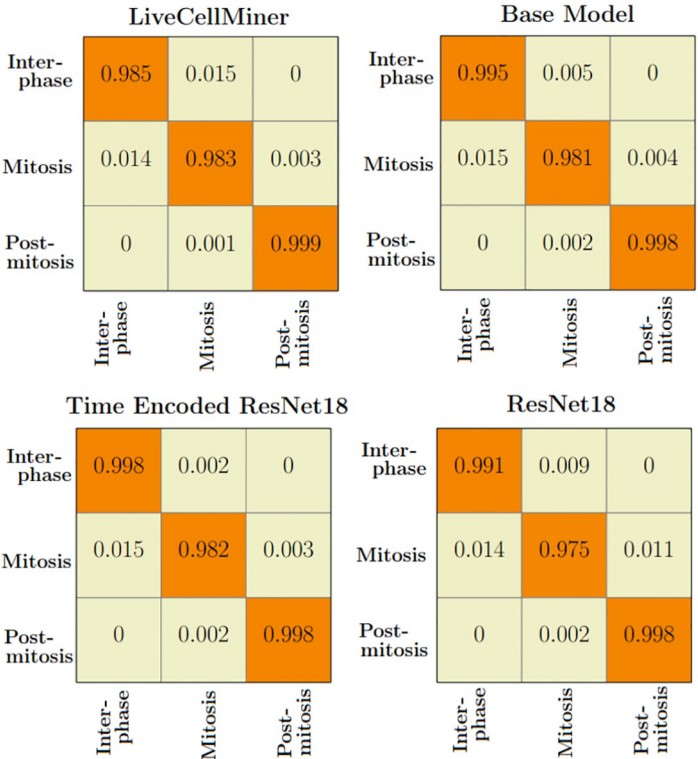

**Fig 7. Confusion matrix plot.** Normalized confusion matrices of the prediction with the LSM710 test dataset.

**Table 3. Average precision, recall, and F-score for the LSM710 dataset.** Average precision, recall, and F-score for each stage of mitosis for the LSM710 dataset. Boldface indicates the best performance.

| Model | Precision | | |
|---|---|---|---|
| | Interphase | Mitosis | Post-mitosis |
| LiveCellMiner | 98.569 | 98.374 | 99.745 |
| Base Model | 99.129±0.207 | 98.302±0.318 | 99.893±0.035 |
| Time Encoded ResNet18 | **99.199±0.19** | **98.479±0.166** | **99.912±0.031** |
| ResNet18 | 99.125±0.061 | 97.563±0.281 | 99.763±0.021 |
| | Recall | | |
| | Interphase | Mitosis | Post-mitosis |
| LiveCellMiner | 98.492 | **98.315** | **99.883** |
| Base Model | 99.46±0.108 | 98.097±0.42 | 99.83±0.038 |
| Time Encoded ResNet18 | **99.769±0.119** | 98.172±0.268 | 99.864±0.032 |
| ResNet18 | 99.182±0.063 | 97.451±0.114 | 99.767±0.062 |
| | F-score | | |
| | Interphase | Mitosis | Post-mitosis |
| LiveCellMiner | 98.531 | **98.344** | 99.814 |
| Base Model | 99.294±0.141 | 98.2±0.351 | 99.862±0.03 |
| Time Encoded ResNet18 | **99.452±0.088** | 98.325±0.167 | **99.898±0.011** |
| ResNet18 | 99.153±0.044 | 97.507±0.137 | 99.765±0.028 |

to the time encoded ResNet18 model. This could be because the embedding space of the base model is at a higher spatial dimension compared to the other model. This tracking network can give the segmentation of the center-cell, with a simple intensity thresholding operation as shown in Fig 8. The embedding space or the feature space gives characteristics related to some properties of the input data. From this feature space in a lower dimension, it can be understood approximately how the classes are divided. Fig 9 illustrates the first three principal components of the feature space of our proposed models. It can be seen that time encoded ResNet18 modules have a proper separation between the features belonging to each class. The higher performance scores of this model could arise because the features belonging to each class are well separated compared to other models. The features of the baseline model also look

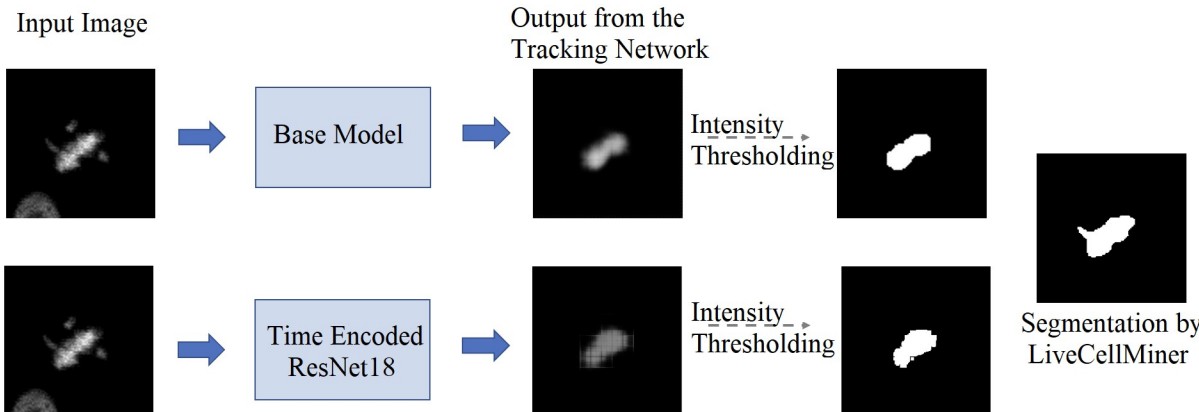

**Fig 8. Tracking network output.** Illustration of the output of the tracking network of the proposed models on the LSM710 dataset. The tracking network reconstructs the center-cell given an input image. Additionally shown the ground-truth mask and the segmentation using intensity thresholding of each output.

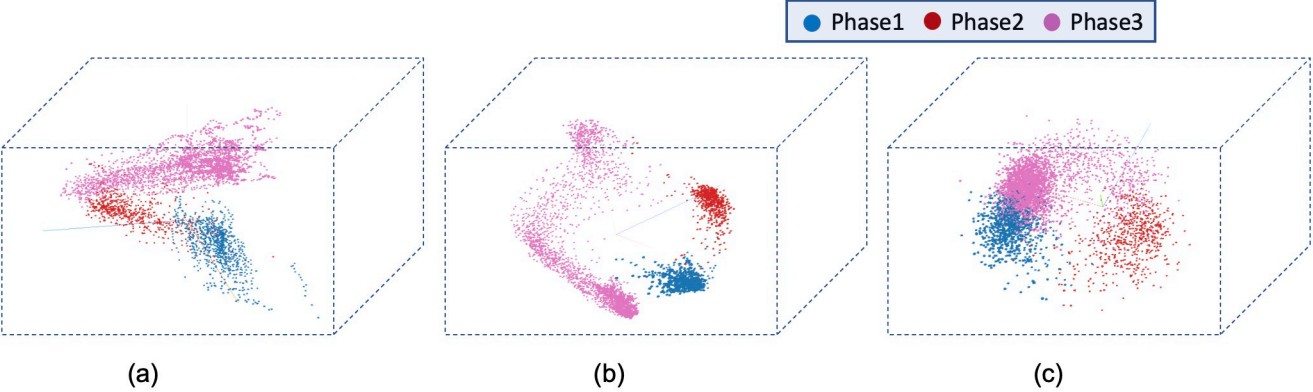

**Fig 9. PCA plot for the images from the LSD1 dataset.** Illustration of the first three principal components of the embeddings of the proposed models. a) Base Model b) Time Encoded ResNet18 c) ResNet18.

well separated. During the postmitotic stage, the daughter cells will grow back to the interphase stage. Since the ResNet18 operates on individual images instead of sequences, less separation between interphase and post-mitosis features is visible. This does not occur in models with GRU layers because the feature space includes the time information which is the main advantage of the proposed model. Hence as seen in Fig 9, time encoded ResNet18 model got better separation of feature embeddings of the classes.

**LSD1 dataset.** The results of the proposed models trained with the LSD1 dataset from LiveCellMiner are presented in this subsection. Fig 10 visualizes the label matrix of the ground truth annotation and the labels generated by the proposed models on 50 test sequences from the LSD1 dataset. Each sequence has a length of 90 frames.

The predicted label matrix of the proposed models compared to the user annotation labels looks similar. This means that each of the four proposed models produced excellent results on this dataset. Further detailed evaluation of the performance is achieved by computing the frame-to-frame accuracy of the predictions on the test dataset. In this dataset, the classification using the ResNet18 model got the least accuracy, and the time encoded ResNet18 model has the maximum performance in terms of accuracy. Thus it seems that introducing features from the previous time step using GRU layers helps in increasing the performance. The precision, recall, and F-score of the three classes of various models are presented in Table 4. Table 5 demonstrates the frame-to-frame accuracy of the proposed models. Except for the precision values in the mitosis and post-mitosis classes, the proposed time encoded ResNet18 model has the highest scores in all other cases. Thus it seems that introducing GRU layers between residual blocks ResNet18 architecture helps in increasing the performance of the model.

The confusion matrix represents the correct predictions as well as the wrong predictions. Fig 11 visualizes the normalized confusion matrices for the proposed models. The diagonal values of the confusion matrix represent the true positive predictions for each class. With the LSD1 dataset, the diagonal elements of the confusion matrix have higher values than the off-diagonal values. This implies that the proposed models were able to classify the images into the correct classes. The PCA plot in Fig 12, indicates better clustering of feature embedding, for the time encoded ResNet18 model compared to the classifier model.

**RecQL4 dataset.** The results of the proposed models trained with the RecQL4 dataset from LiveCellMiner are demonstrated in this subsection. Fig 13 visualizes the label matrix of the ground-truth annotation and the labels generated by the proposed models on test

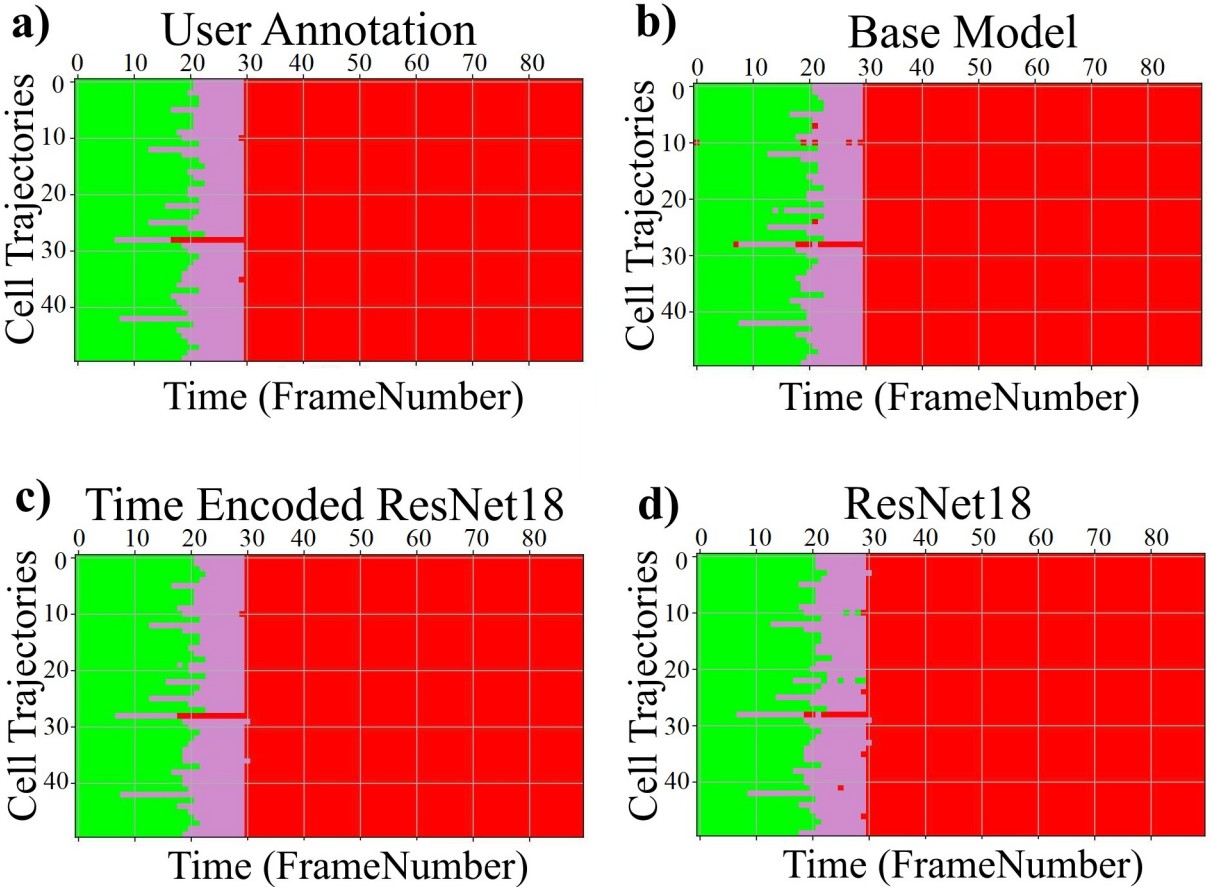

**Fig 10. Label matrix.** a) Label matrices of user annotation and the predictions by proposed models for 50 sequences selected from the test data of the LSD1 dataset. The y-axis represents different cell trajectories, and the x-axis represents the length of each trajectory. Green, magenta, and red represent the interphase, mitosis, and post-mitosis classes respectively.

sequences. The label matrix generated by the ResNet18 model contains some misclassifications in the post-mitosis class and also in the mitosis class for some sequences which is eliminated by the proposed RNN-based model. The same behaviour is observed for the other datasets as well which is shown in the respective subsections. The predicted label matrix of the proposed models compared to the user annotation show a very good correspondence. This indicates that in this dataset, both the proposed models produced good results. By calculating the frame-to-frame accuracy of the predictions on the test dataset, a further assessment of the performance is made possible.

Table 6 demonstrates the frame-to-frame accuracy of each of the proposed models. With the RecQL4 dataset, the base model, and the time encoded ResNet18 model have similar accuracies with a slightly higher value for the time encoded ResNet18 model. The ResNet18 model has the least frame-to-frame accuracy clearly showing the advantage of the proposed model. The precision, recall, and F-score of the three classes of various models for the RecQL4 dataset are presented in Table 7. These values help in the analysis of each model's performance in predicting each class.

It is evident from Table 7 that the time encoded ResNet18 model gave the highest performance scores for most metrics among all models. For some cases, the performance is slightly

**Table 4. Average precision, recall, and F-score for the LSD1 dataset.** Average precision, recall, and F-score for each stage of mitosis for the LSD1 dataset. Boldface indicates the best performance.

| Model | Precision | | |
|---|---|---|---|
| | Interphase | Mitosis | Post-mitosis |
| | Interphase | Mitosis | Post-mitosis |
| LiveCellMiner | 97.837 | **98.549** | **99.989** |
| Base Model | 98.857±0.192 | 98.353±0.131 | 99.896±0.153 |
| Time Encoded ResNet18 | **98.983±0.129** | 98.436±0.187 | 99.957±0.047 |
| ResNet18 | 96.05±0.363 | 98.256±0.317 | 99.622±0.069 |
| *Recall* | Precision | | |
| | Interphase | Mitosis | Post-mitosis |
| LiveCellMiner | 99.539 | 97.792 | 99.018 |
| Base Model | 99.224±0.406 | 97.639±0.463 | 99.802±0.017 |
| Time Encoded ResNet18 | **99.654±0.132** | **97.964±0.252** | **99.823±0.035** |
| ResNet18 | 99.517±0.04 | 92.569±0.697 | 99.404±0.058 |
| *F-score* | Precision | | |
| | Interphase | Mitosis | Post-mitosis |
| LiveCellMiner | 98.681 | 98.169 | 99.501 |
| Base Model | 99.04±0.265 | 97.994±0.288 | 99.899±0.079 |
| Time Encoded ResNet18 | **99.314±0.072** | **98.199±0.097** | **99.902±0.024** |
| ResNet18 | 97.849±0.182 | 95.327±0.441 | 99.513±0.048 |

better for the LiveCellMiner approach. However, the overall performance scores are highest for the time encoded ResNet18 model for most of the cases. The RecQL4 dataset is also annotated into three stages of cell-cycle. The confusion matrix portrays the correct predictions as well as the incorrect predictions. Fig 14 visualizes the normalized confusion matrices for the proposed models. The diagonal values of the confusion matrix represent the true positive predictions for each of the three classes. With the RecQL4 dataset, the diagonal elements of the confusion matrix have higher values than the off-diagonal values. This implies that the proposed models mostly were able to classify the images into correct classes. The PCA plot (Fig 15) has similar behaviour as the other two datasets, with embeddings corresponding to time encoded ResNet18 model showing time continuity as well as clear clustering in the embedding space.

**NikonXLight dataset.** The results of the proposed models trained with the NikonXLight dataset from LiveCellMiner are discussed below. Fig 16 visualizes the label matrix of the ground-truth annotation and the labels generated by the proposed models on test sequences. The predicted label matrix of the proposed models compared to the user annotation labels looks comparable. This indicates that with the NikonXLight dataset, each of the three proposed models produced predictions close to the user annotation. By computing the frame-to-frame

**Table 5. The frame-to-frame accuracy values.** The frame-to-frame accuracy value of various proposed models using the LSD1 dataset. Boldface indicates the best performance.

| Model | Accuracy |
|---|---|
| LiveCellMiner | 98.98 |
| Base Model | 99.508±0.119 |
| Time Encoded ResNet18 | **99.572±0.030** |
| ResNet18 | 98.677±0.103 |

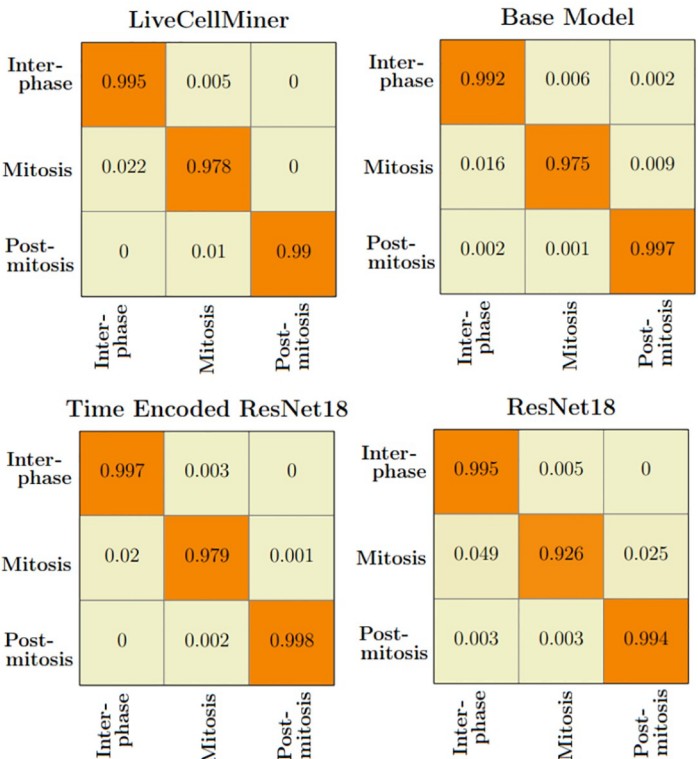

**Fig 11. Confusion matrix plot.** Normalized confusion matrices of the prediction with the LSD1 test dataset.

accuracy of the predictions on the test dataset, a better assessment of the performance is made possible.

Table 8 demonstrates the frame-to-frame accuracy of the proposed models. For the NikonXLight dataset, the time encoded ResNet18 model has slightly lower frame-to-frame accuracy compared to the LiveCellMiner approach. The ResNet18 model has the least frame-to-frame accuracy. This indicates again that introducing the GRU layers to propagate features

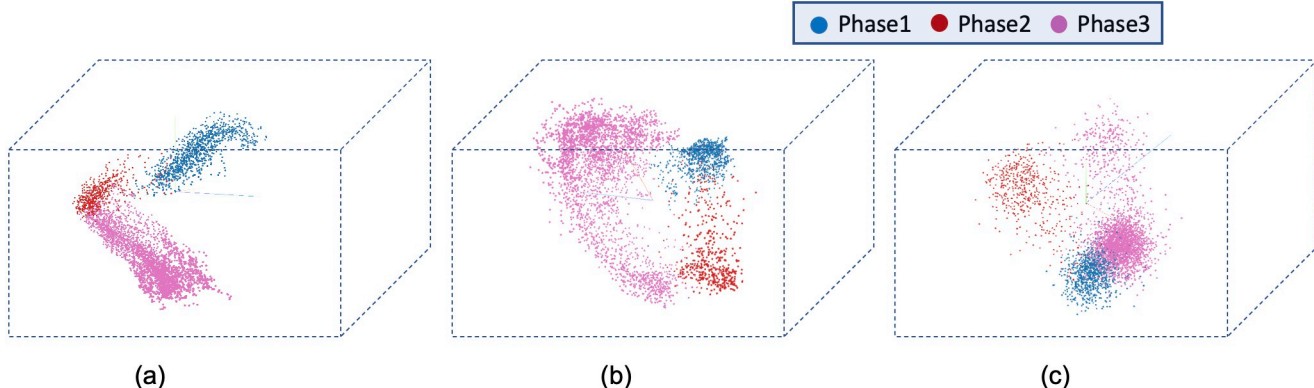

**Fig 12. Illustration of the first three principal components of the embeddings of the proposed models.** a) Base Model b) Time Encoded ResNet18 c) ResNet18.

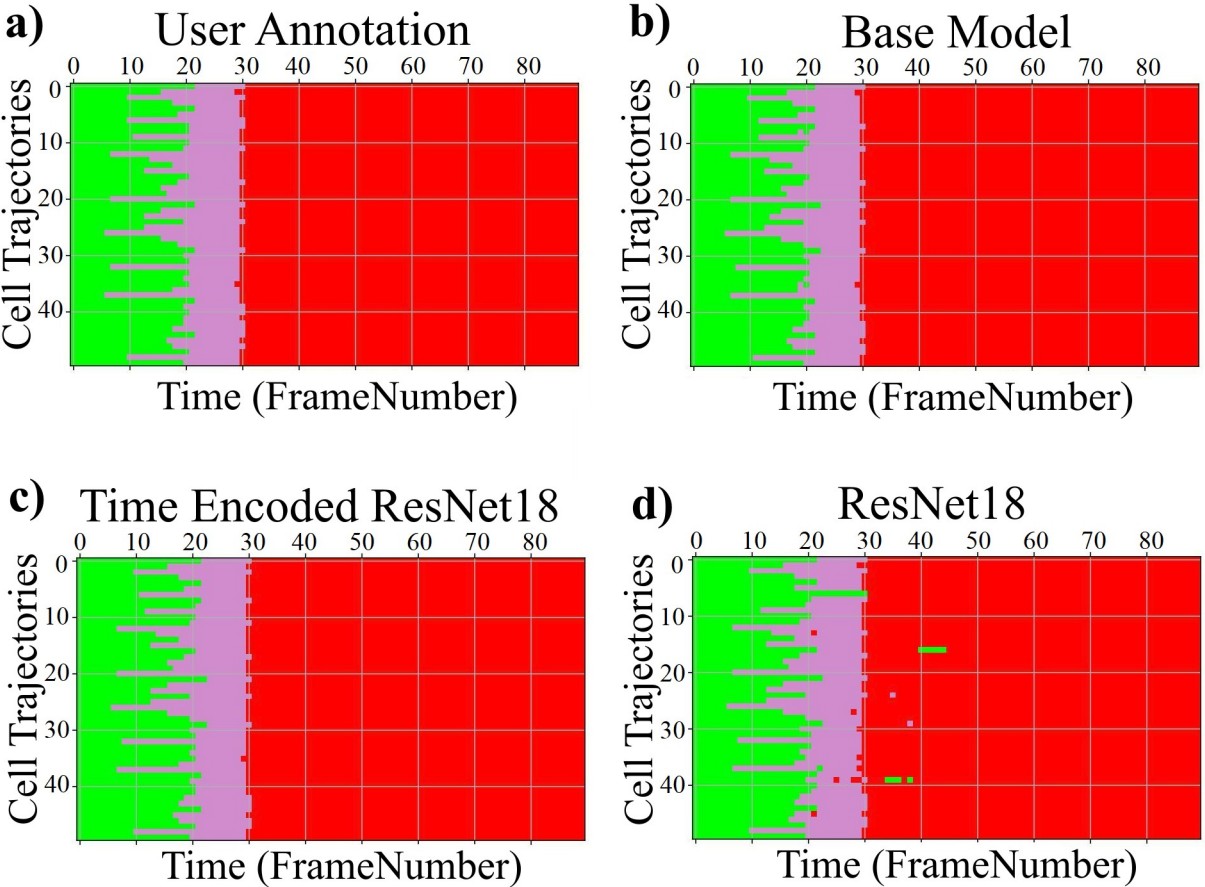

**Fig 13. Label matrix.** Label matrices of user annotation and the predictions by proposed models for 50 sequences selected from the test data of the NikonXLight dataset. The y-axis represents different cell trajectories, and the x-axis represents the length of each trajectory. Green, magenta and red represent the interphase, mitosis, and post-mitosis classes respectively.

with time information between frames in a sequence helps in increasing the frame-to-frame accuracy. The precision, recall, and F-score of the three classes of various models for the NikonXLight dataset are presented in Table 9. These values help in the analysis of each model's performance in predicting each class. It is evident from Table 9 that the time encoded ResNet18 model gave the highest F-scores except for mitosis class, in which LiveCellMiner approach outperforms. The precision and recall values are also highest for the time encoded ResNet18 model in most of the cases, with an exception of precision value in mitosis class and recall value in post-mitosis class in which LiveCellMiner approach slightly outperforms other

**Table 6. The frame-to-frame accuracy values.** The frame-to-frame accuracy values of various proposed models using the RecQL4 dataset. Boldface indicates the best performance.

| Model | Accuracy |
|---|---|
| LiveCellMiner | 98.93 |
| Base Model | 99.322±0.055 |
| Time Encoded ResNet18 | **99.345±0.038** |
| ResNet18 | 98.850±0.114 |

**Table 7. Average precision, recall, and F-score values.** Average precision, recall, and F-score for each stage of mitosis when trained using the RecQL4 dataset on various proposed models. Boldface indicates the best performance.

| *Model* | *Precision* | | |
|---|---|---|---|
| | Interphase | Mitosis | Post-mitosis |
| LiveCellMiner | 98.314 | 98.224 | **99.946** |
| Base Model | 98.465±0.239 | 98.532±0.197 | 99.795±0.1 |
| Time Encoded ResNet18 | **98.856±0.317** | **98.633±0.364** | 99.865±0.119 |
| ResNet18 | 97.101±0.201 | 97.969±0.105 | 99.54±0.12 |
| | *Recall* | | |
| | Interphase | Mitosis | Post-mitosis |
| LiveCellMiner | 99.237 | **98.245** | 98.987 |
| Base Model | 99.346±0.149 | 97.091±0.475 | 99.857±0.032 |
| Time Encoded ResNet18 | **99.422±0.049** | 97.234±0.19 | **99.867±0.299** |
| ResNet18 | 99.197±0.07 | 94.468±0.665 | 99.698±0.057 |
| | *F-score* | | |
| | Interphase | Mitosis | Post-mitosis |
| LiveCellMiner | 98.773 | **98.235** | 99.464 |
| Base Model | 98.903±0.151 | 97.806±0.313 | 99.826±0.06 |
| Time Encoded ResNet18 | **99.124±0.147** | 97.957±0.121 | **99.879±0.049** |
| ResNet18 | 98.138±0.103 | 96.185±0.372 | 99.619±0.069 |

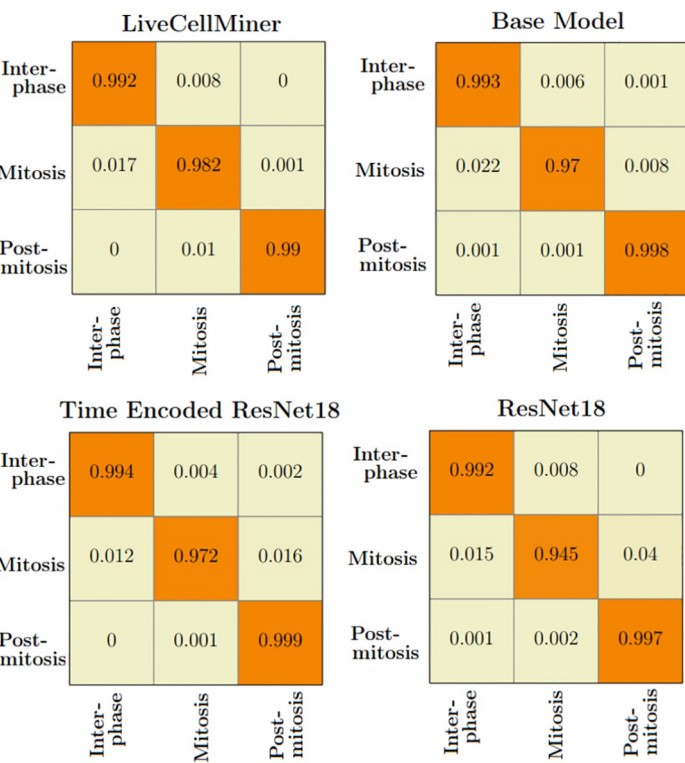

**Fig 14. Normalised confusion matrix.** Normalized confusion matrices of the prediction with the RecQL4 test dataset with proposed models.

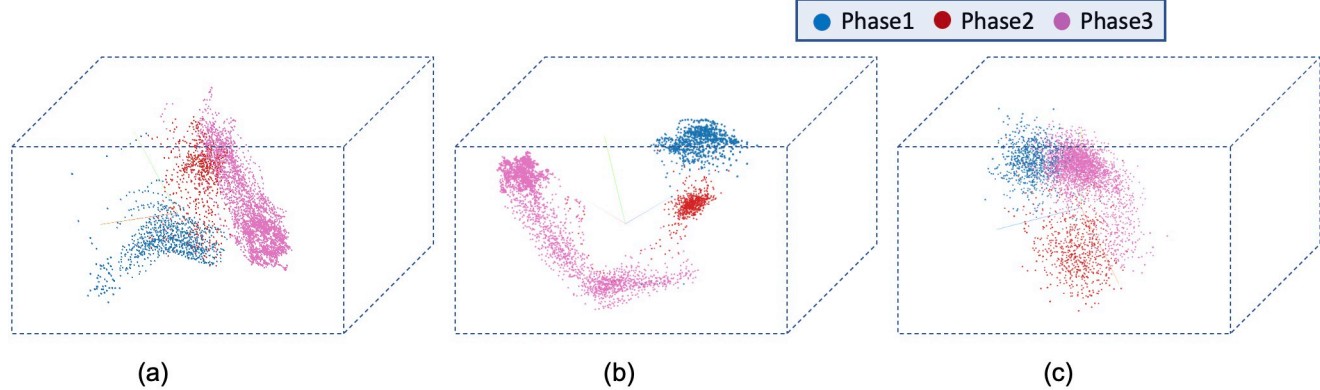

**Fig 15. PCA plot.** Illustration of the first three principal components of the embeddings of the proposed models. a) Base Model b) Time Encoded ResNet18 c) ResNet18.

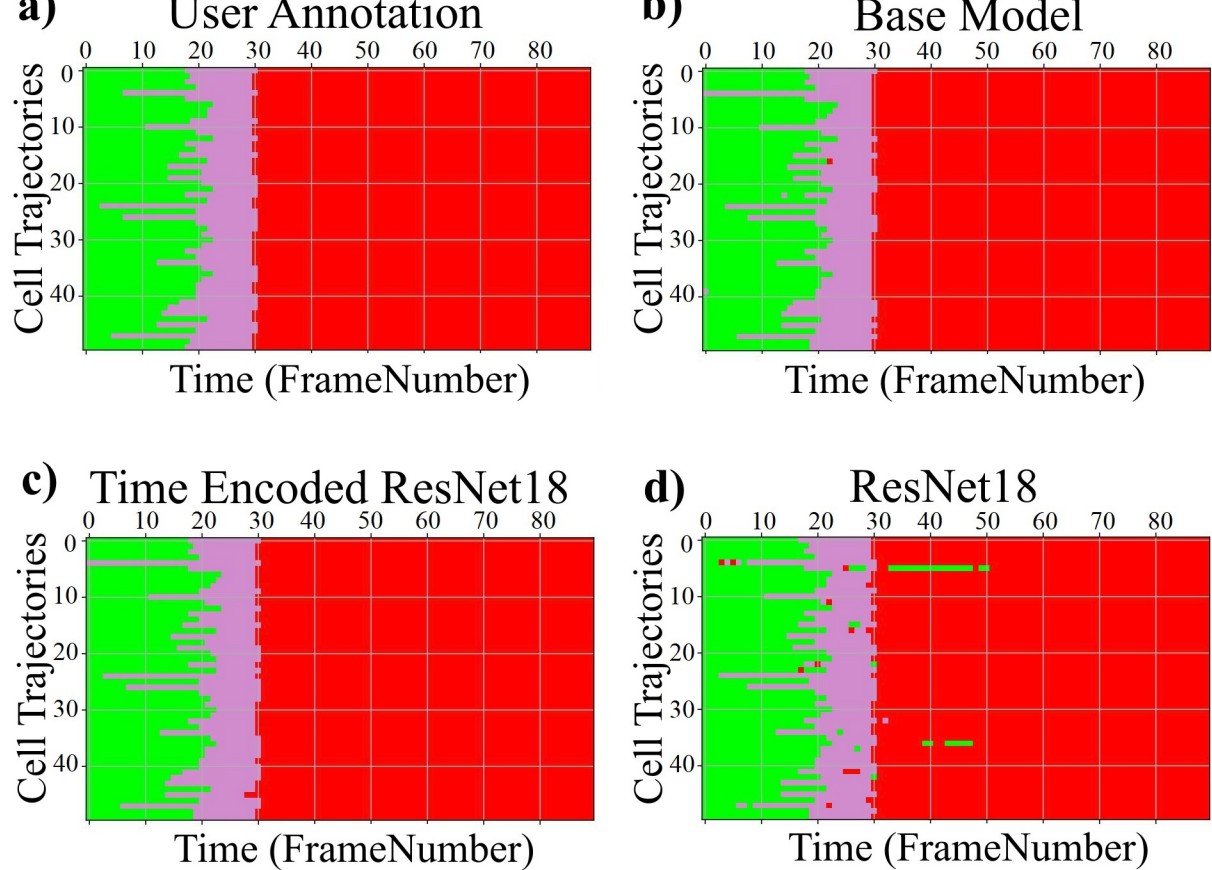

**Fig 16. Label matrix.** Label matrices of user annotation and the predictions by proposed models for 51 sequences selected from the test data of the Zhong et al.'s dataset. The y-axis represents different cell trajectories, and the x-axis represents the length of each trajectory. Green, yellow, orange, violet, blue, and red represent interphase, prophase, prometaphase, metaphase, anaphase, and telophase classes respectively.

**Table 8. The frame-to-frame accuracy values.** The frame-to-frame accuracy value of various proposed models using the NikonXLight dataset. Boldface indicates the best performance.

| Model | Accuracy |
|---|---|
| LiveCellMiner | **99.37** |
| Base Model | 99.269±0.140 |
| Time Encoded ResNet18 | 99.292±0.058 |
| ResNet18 | 98.436±0.127 |

models. Fig 17 visualizes the normalized confusion matrices for the proposed models. The diagonal values of the confusion matrix represent the true positive predictions for each of the three classes. With the NikonXLight dataset, the diagonal elements of the confusion matrix have higher values than the off-diagonal values. This implies that the proposed models mostly were able to classify the images into correct classes. The feature space embeddings shown in Fig 18 shows formation of very compact clusters of different classes for time encoded ResNet18 model, clearly confirming the advantage of the proposed model. We have also measured the classification accuracy of the proposed model in four datasets for two train test ratios, 0.85 and 0.5. With lower number of training data the performance only drops slightly indicating the advantage of the proposed model. Thus, even with less number of annotated data, the network will be able to given reasonably good performance. The results are given in Table 10.

## Results on Zhong et al.'s dataset

This section compares the performance of models with the Zhong et al.'s dataset. As seen in results from the LiveCellMiner dataset, the base model, the time encoded ResNet18 model, and the ResNet18 models are evaluated. Fig 19 visualizes the label matrix of the ground-truth annotation and the labels generated by the proposed models on test sequences. The y-axis in this plot represents the 51 test sequences, and the x-axis is the length of each cell sequence.

**Table 9. Average precision, recall, and F-score values.** Average precision, recall, and F-score for each stage of mitosis when trained using the NikonXLight dataset on various proposed models. Boldface indicates the best performance.

| Model | Precision | | |
|---|---|---|---|
| | Interphase | Mitosis | Post-mitosis |
| LiveCellMiner | 97.229 | **99.190** | 99.009 |
| Base Model | 98.943±0.124 | 97.977±0.377 | 99.624±0.162 |
| Time Encoded ResNet18 | **98.952±0.179** | 98.386±0.412 | **99.671±0.071** |
| ResNet18 | 96.694±0.377 | 97.652±0.22 | 99.138±0.066 |
| Recall | Precision | | |
| | Interphase | Mitosis | Post-mitosis |
| LiveCellMiner | 99.231 | 96.172 | **99.984** |
| Base Model | 99.081±0.131 | 96.666±0.763 | 99.846±0.08 |
| Time Encoded ResNet18 | **99.526±0.093** | **96.723±0.436** | 99.854±0.083 |
| ResNet18 | 99.327±0.084 | 92.269±0.452 | 99.389±0.153 |
| F-score | Precision | | |
| | Interphase | Mitosis | Post-mitosis |
| LiveCellMiner | 98.219 | **97.657** | 99.494 |
| Base Model | 99.012±0.118 | 97.317±0.504 | 99.735±0.113 |
| Time Encoded ResNet18 | **99.233±0.11** | 97.492±0.204 | **99.797±0.026** |
| ResNet18 | 97.992±0.198 | 94.884±0.312 | 99.264±0.099 |

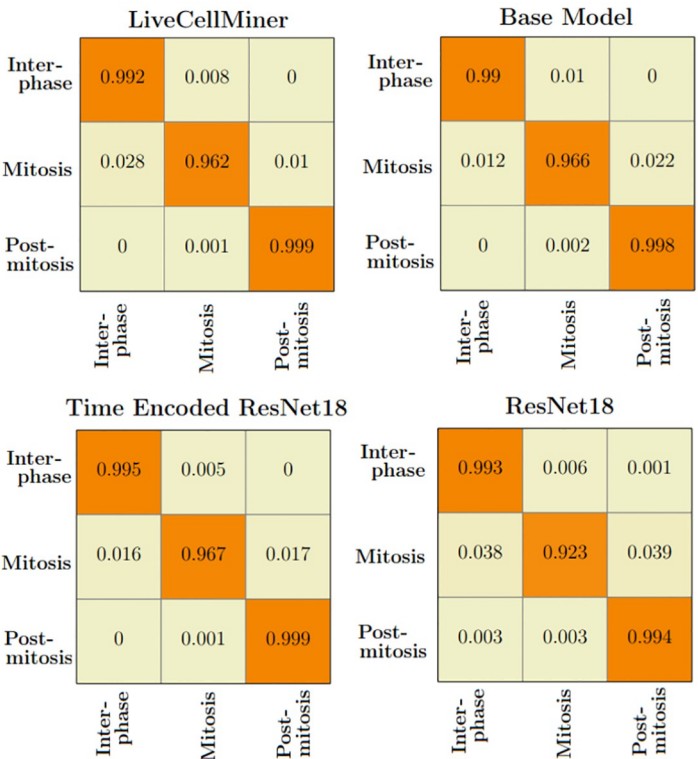

**Fig 17. Normalised confusion matrix.** Normalized confusion matrices of the prediction with the NikonXLight test dataset.

Each cell sequence in this dataset has 40 frames. The predicted label matrix of the proposed models compared to the user annotation labels looks comparable. This indicates that all the models were able to predict closer to ground-truth annotation.

Table 11 shows the precision, recall, and F-score of the proposed models on each class of the Zhong et al.'s dataset. Zhong et al.'s dataset is labeled into six stages of cell-cycle. In Table 11, the first row shows the best model proposed by Zhong et al. [19]. This state-of-the-art method is not a deep learning approach. It is a combination of feature extraction and

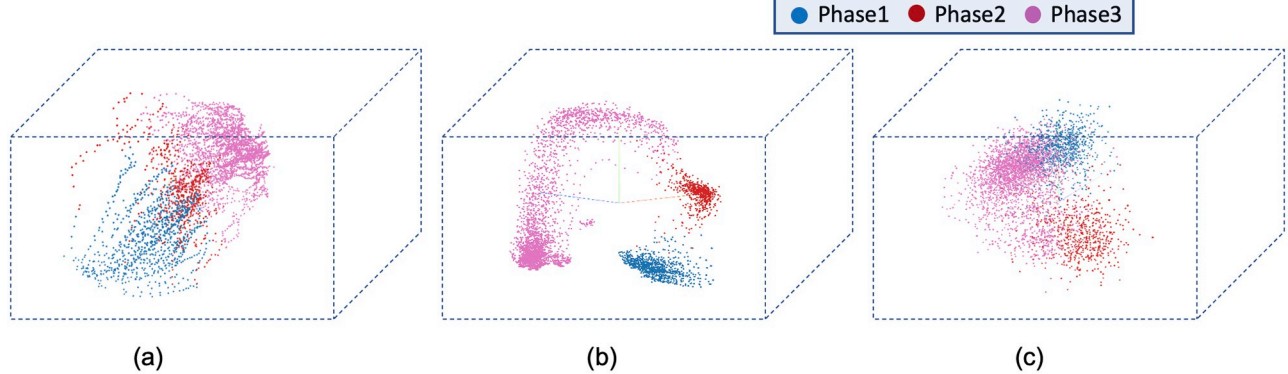

**Fig 18. PCA plot.** Illustration of the first three principal components of the embeddings of the proposed models. a) Base Model b) Time Encoded ResNet18 c) ResNet18.

**Table 10. Classification accuracy for different train test ratios.** Classification accuracy in 4 datasets, for two different train test ratios.

| Dataset | 0.85 | 0.5 |
|---|---|---|
| LSM710 | 99.39 | 96.65 |
| LSD1 | 98.98 | 96.82 |
| RecQL4 | 98.93 | 96.72 |
| NikonXLight | 99.37 | 95.85 |

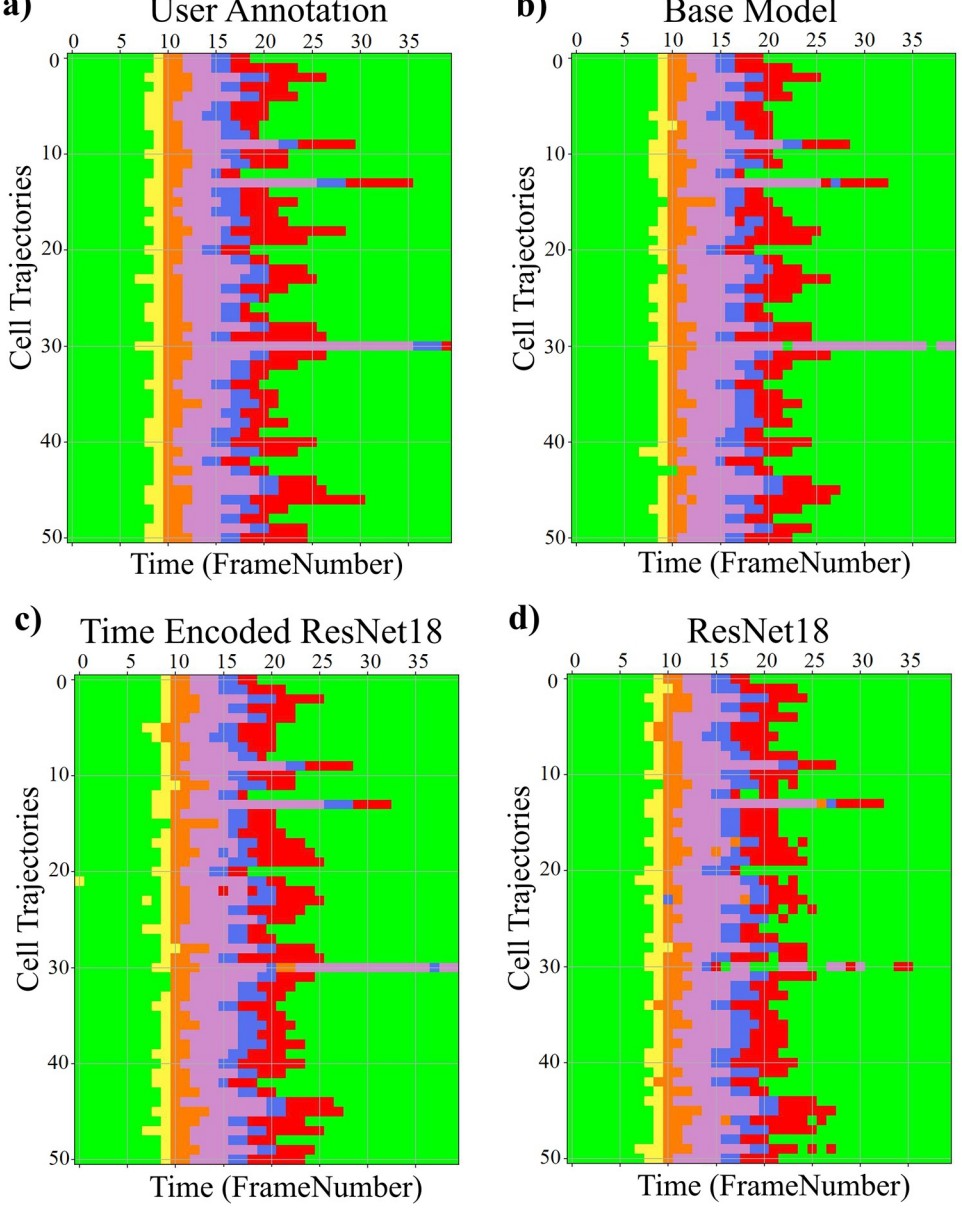

**Fig 19. Label matrix.** Label matrices of user annotation and the predictions by proposed models for 51 sequences selected from the test data of the Zhong et al.'s dataset. The y-axis represents different cell trajectories, and the x-axis represents the length of each trajectory. Green, yellow, orange, violet, blue, and red represent interphase, prophase, prometaphase, metaphase, anaphase, and telophase classes respectively.

**Table 11. Average precision, recall, and F-score.** Average precision, recall, and F-score for each stage of mitosis when trained using the Zhong et al.'s dataset on various proposed models compared to the TC3 model proposed by Zhong et al. [19]. Boldface indicates the best performance.

| Model | Precision | | | | | |
|---|---|---|---|---|---|---|
| | Inter | Pro | Prometa | Meta | Ana | Telo |
| TC3 | 95.97±0.83 | 83.53±2.07 | **91.47±2.45** | **96.82±0.92** | 80.57±7.67 | 84.57±5.28 |
| Base Model | 95.53±0.525 | **88.838±3.035** | 91.004±2.342 | 89.494±1.424 | **91.05±1.057** | **86.763±1.358** |
| Time Encoded | **96.261±0.494** | 85.811±1.566 | 86.729±1.62 | 91.32±1.054 | 90.514±1.531 | 84.507±1.952 |
| ResNet18 | 93.905±0.234 | 83.863±1.211 | 85.751±2.479 | 90.85±0.87 | 86.199±1.965 | 81.959±1.667 |
| | Recall | | | | | |
| | Inter | Pro | Prometa | Meta | Ana | Telo |
| TC3 | **99.51±0.32** | **82.75±4.13** | 84.43±2.96 | 88.24±3.63 | 80.22±6.24 | 79.50±5.09 |
| Base Model | 98.123±0.383 | 71.053±3.419 | 85.613±4.591 | **96.731±1.036** | 79.757±1.496 | 78.807±2.295 |
| Time Encoded | 96.98±0.372 | 81.711±2.387 | **89.34±2.313** | 94.538±0.784 | **85.664±1.89** | **79.727±2.152** |
| ResNet18 | 92.273±0.224 | 80.263±2.372 | 79.835±1.823 | 91.538±0.999 | 81.305±1.359 | 70.568±0.745 |
| | F-score | | | | | |
| | Inter | Pro | Prometa | Meta | Ana | Telo |
| TC3 | **97.69±0.36** | 82.84±2.62 | 87.64±2.35 | 92.05±2.00 | 80.03±6.79 | 81.51±4.70 |
| Base Model | 96.808±0.27 | 78.877±2.311 | **88.163±2.98** | **92.959±0.642** | 85.018±0.914 | **82.571±1.409** |
| Time Encoded | 96.622±0.23 | **83.681±1.338** | 87.95±1.646 | 92.897±0.659 | **87.995±0.818** | 82.015±1.263 |
| ResNet18 | 95.559±0.133 | 82.003±1.425 | 82.668±1.737 | 91.187±0.578 | 83.67±1.357 | 75.829±0.861 |

clustering algorithms. The results of this method compared with results from our proposed models that use deep learning techniques are shown in the table. Our models have slightly lower or higher performance compared to the method proposed by Zhong et al. Within our approaches, the maximum values for each class are distributed across various models. In general, it can be seen that, our proposed models with the GRU layers got higher scores compared to the ResNet18 model. A more precise evaluation can be done by analyzing the frame-to-frame accuracy of each model. Table 12 shows the frame-to-frame accuracy of the proposed models. From this table, it is evident that the time encoded ResNet18 model has the highest frame-to-frame accuracy. So it is clear that our proposed model with GRU layers is able to propagate features between different time instances and increase the performance of the model. The base model which is a very shallow network also has an accuracy higher than the ResNet18 model. The embedding space or the feature space gives characteristics related to some properties of the input data. The first three principal components from principal component analysis (PCA) [20] are used to visualize the embedding space of the proposed models. This is illustrated in Fig 20. The time encoded ResNet18 has the most separate features for each class. This explains the advantage of the proposed model compared to other approaches. The confusion matrix displays the correct predictions as well as the incorrect predictions. Fig 21 visualizes the normalized confusion matrices for the proposed models. The diagonal values

**Table 12. The frame-to-frame accuracy values.** The frame-to-frame accuracy value of various proposed models using the Zhong et al.'s dataset. Boldface indicates the best performance.

| Model | Accuracy |
|---|---|
| TC3 | **94.1** |
| Base Model | 93.142±0.117 |
| Time Encoded ResNet18 | 93.315±0.252 |
| ResNet18 | 91.237±0.273 |

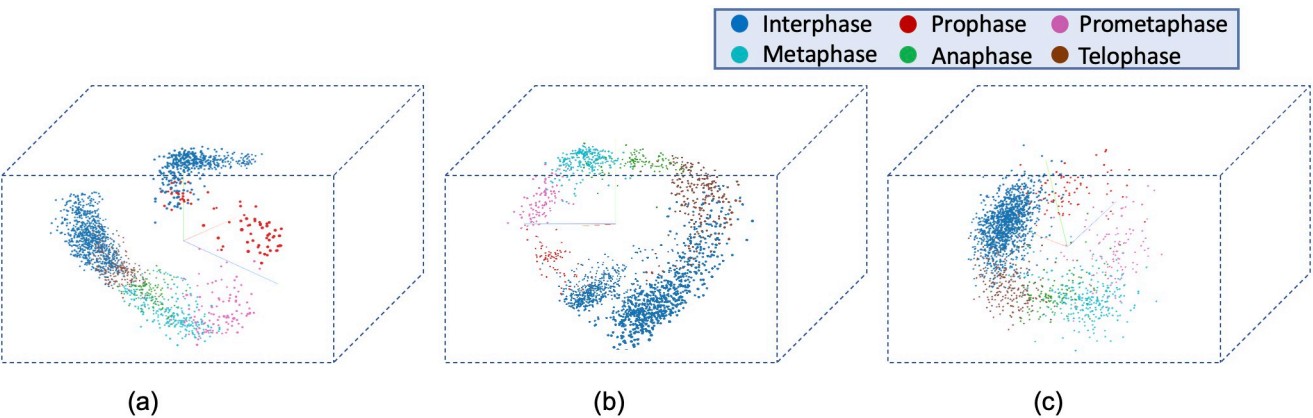

**Fig 20. PCA plot.** Illustration of the first three principal components of the embeddings of the proposed models a) Base Model b) Time Encoded ResNet18 c) ResNet18.

of the confusion matrix represent the true-positive predictions for each of the three classes. For all the proposed models, the diagonal values are higher. This implies that the proposed models are able to predict most of the images into their correct classes. The prophase class has the lowest true-positive rate, which can be attributed to the prophase and interphase cell images having high visual similarities. The true-positive values are higher for the other classes. This denotes that our proposed models were able to give good classification results. We observe that there is a higher rate of misclassification between adjacent classes for the proposed model and the misclassfication value is quite low. This phenomenon comes as an additional advantage stemming from the inclusion of time information. In contrast, for the ResNet18 model without the incorporation of time information, misclassifications are evenly distributed across all classes, and the inclusion of time information mitigates this issue.

## Tracking network results

We have also evaluated the reconstruction results from the network for all the datasets. Figs 22 and 23 shows the reconstruction of center-cell tracked by the proposed models at different stages of cell-splitting for the four LiveCellMiner datasets. The background information from other cells are suppressed in the tracker output. This ensures that the features for classification will not contain any background information. This is because, we train the model with a loss function which calculates the loss between the predicted output and a masked image. Fig 24 shows the output of the tracking network from the proposed models for the Zhong et al.'s

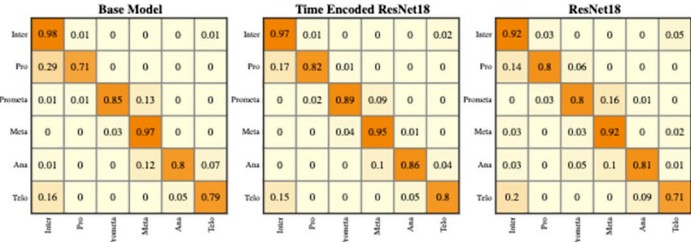

**Fig 21. Normalized confusion matrix.** Normalized confusion matrices of the prediction for the six class dataset.

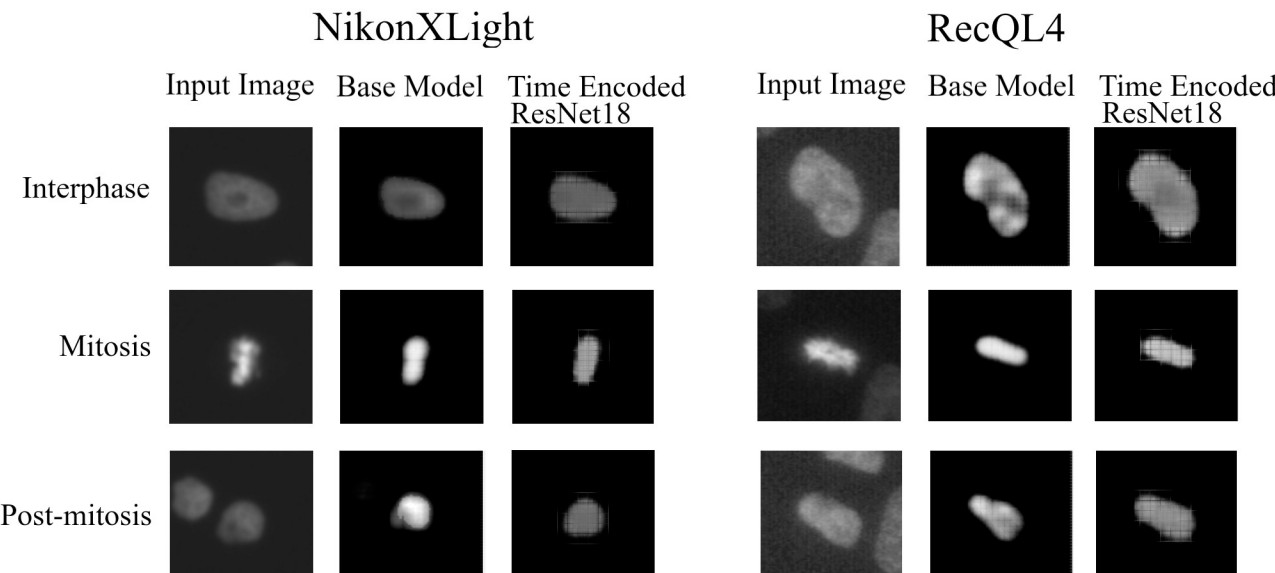

**Fig 22. Reconstruction result of the center-cell.** Center-cell tracking for NikonXLight and RecQL4 datasets.

dataset for six different mitotic phases. Here we could visualize that the neural tracking module of the proposed model is able to track the center-cell, avoid the background clutter from other cells, and extract the features of the cell undergoing mitosis. Training together with the tracker module boosts the performance of the classification model as the feature extraction for the classification happens from the neural features of the tracked cell and background clutter from other cells is completely suppressed, which indeed is the main advantage of our proposed model.

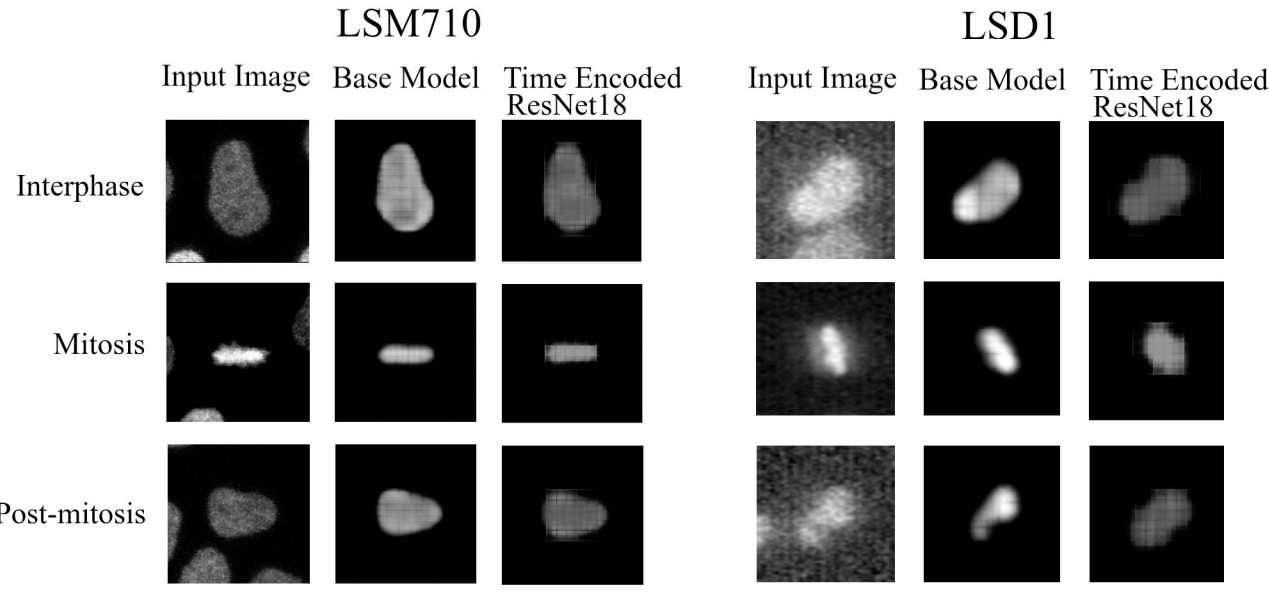

**Fig 23. Reconstruction result of the center-cell.** Center-cell tracking for LSM710 and LSD1 datasets.

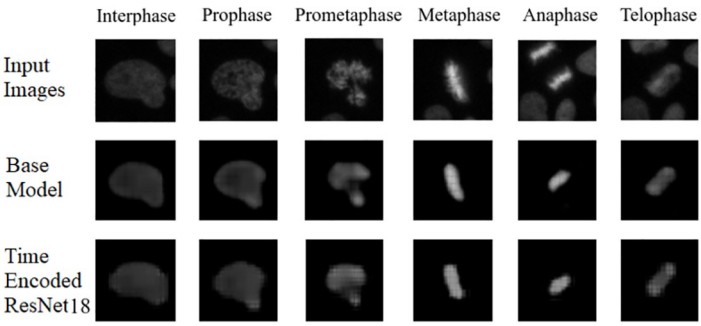

**Fig 24. Tracking network output.** Illustration of the output of the tracking network of the proposed models for input images belonging to different classes. It can be seen that both models were able to reconstruct approximately the center-cell from the input image belonging to various classes. The base model performs the center-cell reconstruction more effectively than other model since the feature space is at a higher dimension.

## Conclusions

In this work, we attempt to study the mitosis process in which the parent cell divides into two identical daughter cells. We also, track the cell division process automatically using deep learning methods. For training the neural network models, we have extracted, video sequences of the cells that undergo cell division and then trained an RNN model, to extract the feature vectors. The use of this RNN has proven to help in better extracting the feature vectors, compared to the conventional classifier models. We have carried out experiments to visualize the feature space to better understand why RNNs are better at identifying the mitosis states. This, in turn, showed that the feature space has a time continuity in the high-dimensional space and clustering is happening as well when trained using RNN model. In addition, we have measured the precision, recall, and F-score as well. This indicates the superiority of the proposed method compared to the ResNet18 baseline. By plotting the confusion matrix, we quantified the amount of misclassification in adjacent classes for both the 3-class and 6-class datasets. Furthermore, the reconstruction results were evaluated to understand how the neural network reconstructs the center-cell during training. Since the center cell is only reconstructed, the network is easily able to extract the features from the cell undergoing cell division, and suppress the clutter from the background cells. This is because during training, the loss is calculated between the predicted output and a masked image. A notable drawback of the proposed method is its inconsistency in outperforming the state-of-the-art across all phases of mitosis. In experiments with the 6-class dataset, the proposed method achieves a higher F-score only for the prophase and anaphase. However, for the other phases, the numbers are comparable to the state-of-the-art model. Similarly, in the case of the 3-class dataset, the F-score for the mitotic phase is lower. A possible reason for this might be the lower number of frames in this phase for training. As future work, we plan to investigate frame interpolation techniques to increase the number of data frames in this class to mitigate this issue.

## Author Contributions

**Conceptualization:** Abin Jose, Johannes Stegmaier.

**Data curation:** Rijo Roy.

**Formal analysis:** Abin Jose, Rijo Roy.

**Funding acquisition:** Abin Jose, Daniel Moreno-Andrés, Johannes Stegmaier.

**Investigation:** Abin Jose, Rijo Roy.

**Methodology:** Abin Jose, Rijo Roy.

**Project administration:** Abin Jose, Johannes Stegmaier.

**Resources:** Abin Jose, Daniel Moreno-Andrés, Johannes Stegmaier.

**Software:** Abin Jose, Rijo Roy.

**Supervision:** Abin Jose, Johannes Stegmaier.

**Validation:** Abin Jose, Rijo Roy, Daniel Moreno-Andrés.

**Visualization:** Abin Jose, Rijo Roy, Daniel Moreno-Andrés, Johannes Stegmaier.

**Writing – original draft:** Abin Jose, Rijo Roy.

**Writing – review & editing:** Abin Jose, Daniel Moreno-Andrés, Johannes Stegmaier.

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
