## [Decision Letter · Decision Letter 0]

13 Sep 2023

PONE-D-23-27382Automatic Detection of Cell-cycle Stages using Recurrent

Neural NetworksPLOS ONE

Dear Dr. Jose,

Thank you for submitting your manuscript to PLOS ONE. After careful consideration, we feel that it has merit but does not fully meet PLOS ONE’s publication criteria as it currently stands. Therefore, we invite you to submit a revised version of the manuscript that addresses the points raised during the review process.

We look forward to receiving your revised manuscript.

Kind regards,

Xiao Luo

Academic Editor

PLOS ONE

Journal Requirements:

4. We noted in your submission details that a portion of your manuscript may have been presented or published elsewhere. 

"Yes, Fig. 1. was published in ICASSP 2023.This is a pre-work of this Journal paper. The Journal paper covers advanced model architecture and more experiments." 

Please clarify whether this [publication] was peer-reviewed and formally published. If this work was previously peer-reviewed and published, in the cover letter please provide the reason that this work does not constitute dual publication and should be included in the current manuscript.

Reviewers' comments:

Reviewer's Responses to Questions

**Comments to the Author**

1. Is the manuscript technically sound, and do the data support the conclusions?

Reviewer #1: Yes

Reviewer #2: Yes

Reviewer #3: Yes

2. Has the statistical analysis been performed appropriately and rigorously? 

Reviewer #1: Yes

Reviewer #2: Yes

Reviewer #3: N/A

3. Have the authors made all data underlying the findings in their manuscript fully available?

Reviewer #1: Yes

Reviewer #2: Yes

Reviewer #3: No

4. Is the manuscript presented in an intelligible fashion and written in standard English?

Reviewer #1: Yes

Reviewer #2: Yes

Reviewer #3: Yes

5. Review Comments to the Author

Reviewer #1: The authors propose a novel network architecture, called Time Encoded ResNet18 Model, and conduct a series of experiments. Here, i have some questions that need to be answered.

1. Did the authors use any statistical methods at any points in the manuscript? Please updated and include the details in the respective statistical analysis section.

2. Whether the change of hyperparameters will have a drastic effect on the model's performance is hoped that the authors can prove it experimentally.

3.From the performance of label matrix, the superiority of the model proposed by authors seems not obvious, and it is hoped that this results can be further explained.

4. The baseline proposed by authors is relatively few, and it is hoped that some baseline other than RNN-based can be added.

Reviewer #2: Jose et al. proposed a novel deep-learning model named Time Encoded ResNet18 to distinguish different mitotic stage. They evaluated the model by comparing it with the base model and other algorithms and proved the overall superior performance. In real datasets, Time Encoded ResNet18 achieved high accuracy and precision. I have some concerns listed below.

Major:

1. The accuracies, precisions, recalls, and F-scores showed in Table 3, 5, 7, 9, and 11 did not prove the consistent superior performance of Time Encoded ResNet18. In Table 11, the model reached the top F-score in only 2/6 states, which showed that in some cases, the model could not even outperform the base model. Since the evaluation criteria were quite close among four methods (close to 1.000), I didn’t see the advantage of Time Encoded ResNet18.

2. Fig13 shows the predictions of each method and the true labels. But I can hardly tell which method was better. Authors may compare the key difference of each method and plot the figures focusing on the difference.

Minor:

3. Please define the term “sequence” clearly.

4. Please explain why models treat each image independently in line 423.

5. The typo in equation 8.

Reviewer #3: In the paper, the authors developed a deep-learning framework to predict mitosis stages from time-lapse microscopy images. They adopted an RNN to capture the connections between successive frames. With several in-depth experiments on two public datasets, they demonstrated that their methods performed better than the baseline methods. Besides, they also studied the features extracted by different models and explained why their RNN model had superior performance.

Strength:

1. Instead of viewing each frame as an independent sample, the authors use RNN to capture the time dependency between successive frames.

2. Features extracted by their method could clearly distinguish different mitosis stages.

3. The tracking network in their model can automatically track the center-cell and suppress information from other cells, which brings convenience for users.

Weakness:

1. In their literature review, the authors mentioned a state-of-the-art methods to identify cell-cycle stages, CellCognition and LiveCellMiner. However, they did not compare the performance of their model with CellCognition and only compared with LiverCellMiner on the "LiveCellMiner Dataset". In my point of view, other competing methods they adopted were more like an ablation study. Therefore, I suggest adding some experiments to compare the performance of their Time Encoded ResNet18 with CellCognition and LiveCellMiner on both the "LiveCellMiner Dataset" and the "Zhong et al.’s Dataset".

2. In their experiments, the training and testing were performed within the same dataset. However, in practice, after getting some new time-lapse microscopy images, it is usually time-consuming to annotate some of them to get a training set. Therefore, I suggest the authors to test the performance of their model when training on one dataset, say, the LSM710 dataset and predicting the cell-cycle stages of another dataset containing the same type of cells (maybe after fine-tuning on a small fraction of data), say, the LSD1 dataset (also capturing the human HeLa cells). Or, maybe they can test their model on one dataset they used but with a smaller train-test ratio, say, 0.5 or even 0.2. I think these settings will make their model more attractive and helpful for users.

3. It seems that the improvements in accuracy, recall and F-score were quite marginal in some experiments, for example, Table 2, Table 8 and Table 10. Also, I noticed that in several settings, their Time Encoded ResNet18 had a lower F-score than LiveCellMiner for the identification of cells belonging to the mitosis stage (Table 3, Table 7, Table 9). Therefore, I was wondering if the improvements in classification brought by their model can help obtain some biological findings. For example, the authors of LiveCellMiner [1] analyzed the NikonXLight dataset and found that downregulation of INO80, SRCAP, EP400 and H2A.Z consistently lengthens early mitotic progression. They also detailedly analyzed the LSD1 dataset and performed some experiments to support their findings. Hence, I suggest the authors carefully examine the results of at least one of the datasets in their experiments and explain or discuss whether any possible new biological findings can be found when applying their Time Encoded ResNet18 model.

4. In the "Conclusions" part, the authors did not discuss the limitations of their model, which are important for users to correctly use their model and avoid possible pitfalls.

Minor points:

1. The authors did not provide their codes for this model, making it difficult to try their methods.

2. In Table 8, the accuracy of LiveCellMiner was higher than the mean accuracy of the Time Encoded ResNet18 model. Therefore, LiveCellMiner should be in boldface rather than Time Encoded ResNet18.

3. In Table 10 (the analysis of Zhong et al.’s Dataset), I suggest adding the frame-to-frame accuracy of TC3, since it performed the best in predicting cells belonging to the interphase.

4. In the last line on page 3, the names of the authors of reference [26] were missing (after the words "in 2016").

References:

[1] Moreno-Andr´es D, Bhattacharyya A, Scheufen A, Stegmaier J. LiveCellMiner: A new tool to analyze mitotic progression. PloS one. 2022;17(7):e0270923.

6. PLOS authors have the option to publish the peer review history of their article (what does this mean?). If published, this will include your full peer review and any attached files.

Reviewer #1: No

Reviewer #2: No

Reviewer #3: No

---

## [Author Response · Author response to Decision Letter 0]

17 Dec 2023

Response to the editor. 

We have updated this section.

3. In your Data Availability statement : 

Code is now available in the repo : https://github.com/Rijo756/cell-cycle-stages-identification. Dataset will be updated soon. We are working on it now.

4. We noted in your submission details that a portion of your manuscript may have been presented or published elsewhere. 

"Yes, Fig. 1. was published in ICASSP 2023. This is a pre-work of this Journal paper. The Journal paper covers advanced model architecture and more experiments." 

Please clarify whether this [publication] was peer-reviewed and formally published. We have mentioned this in the cover letter, that only the Fig. 1 is resued.

Response to the reviewers.

We thank the reviewers for their critical assessment of our approach. In the following, we address

the major concerns raised point by point, and the corresponding revisions.

Comments from Reviewer 1

Comment: 1. Did the authors use any statistical methods at any points in the manuscript? Please

update and include the details in the respective statistical analysis section.

Reply: Thanks for your comment. Kindly note that the reported numbers in the paper are the average

performance of 20 repetitions and we have included the standard deviations of each experiment as well

in the tables. Please check line number 447.

Comment: 2. Whether the change of hyperparameters will have a drastic effect on the model's

performance is hoped that the authors can prove it experimentally.

Reply: Thanks for the comment. The results reported have already undergone grid search-based

hyperparameter tuning. The optimum hyperparameters selected by the grid search are provided in

Table 1, and the standard deviation of 20 repetitions is already mentioned in the paper. Please check

line 447 and 448 in the manuscript.

Comment: 3. From the performance of label matrix, the superiority of the model proposed by

authors seems not obvious, and it is hoped that this results can be further explained.

Reply: It is true that visually \fnding the differences in the label matrix can be challenging. To address

this, we have increased the size of the label matrix plots to make the differences more clear. Please

refer to Fig. 6, Fig. 7, Fig. 10, Fig. 11, and Fig. 13 for visual representations. Additionally, we have

included quantitative metrics such as precision, recall, and F-score. To further illustrate the advantages

of incorporating time information, we have included the PCA plots. Furthermore, the accuracy, which

is a qualitative measure of the label matrix, is clearly demonstrated in the accuracy table, highlighting

the advantages of the proposed method. We have further added the following sentence in the results

section. "The label matrix generated by the ResNet18 model contains some misclassi\fcations in the

post-mitosis class and also in the mitosis class for some sequences which is eliminated by the proposed

RNN-based model. The same behaviour is observed for the other datasets as well which is shown in the

respective subsections." (Please refer to line numbers 552 to line 555 in the manuscript).

Comment: 4. The baseline proposed by authors is relatively few, and it is hoped that some

baseline other than RNN-based can be added.

Reply: Kindly note that, we have included the ResNet18 model which is not an RNN-based approach

for comparison. We have also compared our model with LiveCellMiner paper and the baseline model

we have developed. We are con\fdent that the proposed baseline re

ects the state-of-the-art.

Comments from Reviewer 2

1

Comment: 1. The accuracies, precisions, recalls, and F-scores showed in Table 3, 5, 7, 9, and

11 did not prove the consistent superior performance of Time Encoded ResNet18. In Table 11,

the model reached the top F-score in only 2/6 states, which showed that in some cases, the model

could not even outperform the base model. Since the evaluation criteria were quite close among

four methods (close to 1.000), I didn't see the advantage of Time Encoded ResNet18.

Reply: Kindly note that, even though the results for the Inter, Prometa, Meta, and Telo phases are

somewhat lower, they are quite close to those of the best-performing model. In the two states where

the results are better, we observe that the numbers are signi\fcantly higher compared to the second

best-performing model. Also please refer to the Tables 3, 7, and 9 in the results section. It is quite

clear that the results are only lower for mitosis state and for the other two states, the F-score is already

higher.

Comment: 2. Fig. 13 shows the predictions of each method and the true labels. But I can hardly

tell which method was better. Authors may compare the key difference of each method and plot

the \fgures focusing on the difference.

Reply: Thanks for pointing out this. It is true that label matrix plots were small. Now we have

increased the size of the plots and updated the label matrix plots and the latent space plots. Kindly

refer to the response to comment 3 of reviewer 1 in the rebuttal letter.

Minor comments:

Comment: 3. Please de\fne the term \\sequence" clearly.

Reply: Kindly note that a "Sequence" typically refers to a series of related or connected frames. We

have indicated that now in the introduction. Please refer to line 13 in the manuscript.

Comment: 4. Please explain why models treat each image independently in line 423.

Reply: The image is treated independently here as it is a classi\fer model and time information is not

used. Please refer to line 422 in the manuscript.

Comment: 5. The typo in equation 8.

Reply: We have now corrected the typo in equation 8. Please refer to line 418.

Comments from Reviewer 3

Comment: 1. In their literature review, the authors mentioned a state-of-the-art methods to

identify cell-cycle stages, CellCognition and LiveCellMiner. However, they did not compare the

performance of their model with CellCognition and only compared with LiverCellMiner on the

"LiveCellMiner Dataset". In my point of view, other competing methods they adopted were more

like an ablation study. Therefore, I suggest adding some experiments to compare the performance

of their Time Encoded ResNet18 with CellCognition and LiveCellMiner on both the "LiveCellMiner

Dataset" and the "Zhong et al.'s Dataset".

2

Reply: CellCognition is an annotation tool used for \fnding complex cellular dynamics. It uses a whole-

slide 2D+t image which contains multiple cells to detect objects and classify them into cell-cycle stages.

However, in our case, both the datasets are single cell datasets which contain only one cell. Since our

focus is on single cell image datasets, it is not reasonable to compare with CellCognition. Furthermore,

the Zhong et al.'s dataset even uses CellCognition for object detection and feature extraction from the

whole-slide images. Kindly note that, the comparison results of the performance of the Time Encoded

ResNet18 with LiveCellMiner and TC3 are already available.

Comment: 2. In their experiments, the training and testing were performed within the same

dataset. However, in practice, after getting some new time-lapse microscopy images, it is usually

time-consuming to annotate some of them to get a training set. Therefore, I suggest the authors

to test the performance of their model when training on one dataset, say, the LSM710 dataset and

predicting the cell-cycle stages of another dataset containing the same type of cells (maybe after

\fne-tuning on a small fraction of data), say, the LSD1 dataset (also capturing the human HeLa

cells). Or, maybe they can test their model on one dataset they used but with a smaller train-test

ratio, say, 0.5 or even 0.2. I think these settings will make their model more attractive and helpful

for users.

Reply: Thanks for the comment. We guess the performance will drop quite a bit if we change the

modality. The proposed experiment of checking what fraction of ground truth is needed to obtain good

results would, however, be quite interesting and also a hint for potential users how many images they

should annotate to obtain good results. We have done the experiment for train test ratio of 0.5 and

the results are now updated in Table 10. It shows that the numbers do not drop signi\fcantly indicating

that even with a smaller dataset, the model is able to give good performance.

Comment: 3. It seems that the improvements in accuracy, recall and F-score were quite marginal

in some experiments, for example, Table 2, Table 8 and Table 10. Also, I noticed that in several set-

tings, their Time Encoded ResNet18 had a lower F-score than LiveCellMiner for the identi\fcation

of cells belonging to the mitosis stage (Table 3, Table 7, Table 9). Therefore, I was wondering if

the improvements in classi\fcation brought by their model can help obtain some biological \fndings.

For example, the authors of LiveCellMiner [1] analyzed the NikonXLight dataset and found that

downregulation of INO80, SRCAP, EP400 and H2A.Z consistently lengthens early mitotic progres-

sion. They also detailedly analyzed the LSD1 dataset and performed some experiments to support

their \fndings. Hence, I suggest the authors carefully examine the results of at least one of the

datasets in their experiments and explain or discuss whether any possible new biological \fndings

can be found when applying their Time Encoded ResNet18 model.

Reply: Thank you for your comment. We were not able to make any new biological conclusions from

the experiments. However, in our experiments with the 6-class dataset, we observed that there is a

higher rate of misclassi\fcation between adjacent classes in the case of the time-encoded ResNet18 and

the misclassi\fcation value is quite low for the other classes. This phenomenon comes as an additional

advantage stemming from the inclusion of time information. In contrast, for the ResNet18 model

without the incorporation of time information, misclassi\fcations are distributed more evenly across all

classes. We have added this information now in the paper. Please refer to line numbers 654-659 in the

manuscript.

3

Comment: 4. In the "Conclusions" part, the authors did not discuss the limitations of their

model, which are important for users to correctly use their model and avoid possible pitfalls.

Reply: We have updated the 'Conclusions' section of the paper and included the limitations of the

paper and future work. Please refer to line numbers 690-702.

Minor points:

Comment: 5. The authors did not provide their codes for this model, making it difficult to try

their methods.

Reply: The github link (https://github.com/Rijo756/cell-cycle-stages-identi\fcation) is added now.

(See the footnote in page 13.).

Comment: 6. In Table 8, the accuracy of LiveCellMiner was higher than the mean accuracy of

the Time Encoded ResNet18 model. Therefore, LiveCellMiner should be in boldface rather than

Time Encoded ResNet18.

Reply: Thanks for noting this mistake. We have updated Table 8.

Comment: 7. In Table 10 (the analysis of Zhong et al.'s Dataset), I suggest adding the frame-to-

frame accuracy of TC3, since it performed the best in predicting cells belonging to the interphase.

Reply: We have referred the TC3 [19] paper for the accuracy numbers. The accuracy value is not

reported as a number, but the supplementary \fle contains a graph showing how the number of features

affect the accuracy and the highest value reported for the TC3 model in the paper is 94:1%. We have

added that now in the manuscript. Please refer to Table 11.

It is true that the accuracy is slightly lower for the proposed method. Also, please notice that for the

pro, prometa, meta, anaphase, and telo-phases the F-score is higher than the TC3 approach. These

phases occur much more rarely than the interphase and it is thus great that our method performs better

here.

Comment: 8. In the last line on page 3, the names of the authors of reference [26] were missing

(after the words "in 2016").

Reply: Thanks for noting this typo. We have updated it. Please check line 113.

---

## [Decision Letter · Decision Letter 1]

4 Jan 2024

Automatic Detection of Cell-cycle Stages using Recurrent

Neural Networks

PONE-D-23-27382R1

Dear Dr. Abin Jose,

We’re pleased to inform you that your manuscript has been judged scientifically suitable for publication and will be formally accepted for publication once it meets all outstanding technical requirements.

Kind regards,

Xiao Luo

Academic Editor

PLOS ONE

Additional Editor Comments (optional):

Reviewers' comments:

Reviewer's Responses to Questions

**Comments to the Author**

1. If the authors have adequately addressed your comments raised in a previous round of review and you feel that this manuscript is now acceptable for publication, you may indicate that here to bypass the “Comments to the Author” section, enter your conflict of interest statement in the “Confidential to Editor” section, and submit your "Accept" recommendation.

Reviewer #2: All comments have been addressed

Reviewer #3: All comments have been addressed

2. Is the manuscript technically sound, and do the data support the conclusions?

Reviewer #2: Yes

Reviewer #3: Yes

3. Has the statistical analysis been performed appropriately and rigorously? 

Reviewer #2: Yes

Reviewer #3: N/A

4. Have the authors made all data underlying the findings in their manuscript fully available?

Reviewer #2: Yes

Reviewer #3: Yes

5. Is the manuscript presented in an intelligible fashion and written in standard English?

Reviewer #2: Yes

Reviewer #3: Yes

6. Review Comments to the Author

Reviewer #2: All my concerns were addressed. Nice work! No more comments.

All my concerns were addressed. Nice work! No more comments. (Twice for meeting the character count requirement)

Reviewer #3: I thank the authors for their detailed responses of all the reviewer comments. I have no further concerns and feel that it is now suitable for being published.

7. PLOS authors have the option to publish the peer review history of their article (what does this mean?). If published, this will include your full peer review and any attached files.

Reviewer #2: No

Reviewer #3: No

---

## [Editor Report · Acceptance letter]

15 Feb 2024

PONE-D-23-27382R1 

PLOS ONE

Dear Dr. Jose, 

I'm pleased to inform you that your manuscript has been deemed suitable for publication in PLOS ONE. Congratulations! Your manuscript is now being handed over to our production team.

Kind regards, 

on behalf of

Dr. Xiao Luo 

Academic Editor

PLOS ONE